# Componential Prompt-Knowledge Alignment for Domain Incremental Learning

**Kunlun Xu** [1]  **Xu Zou** [2]  **Gang Hua** [3]  **Jiahuan Zhou*** [1]

## Abstract

Domain Incremental Learning (DIL) aims to learn from non-stationary data streams across domains while retaining and utilizing past knowledge. Although prompt-based methods effectively store multi-domain knowledge in prompt parameters and obtain advanced performance through cross-domain prompt fusion, we reveal an intrinsic limitation: component-wise misalignment between domain-specific prompts leads to conflicting knowledge integration and degraded predictions. This arises from the random positioning of knowledge components within prompts, where irrelevant component fusion introduces interference. To address this, we propose Componential Prompt-Knowledge Alignment (KA-Prompt), a novel prompt-based DIL method that introduces component-aware prompt-knowledge alignment during training, significantly improving both the learning and inference capacity of the model. KA-Prompt operates in two phases: (1) Initial Componential Structure Configuring, where a set of old prompts containing knowledge relevant to the new domain are mined via greedy search, which is then exploited to initialize new prompts to achieve reusable knowledge transfer and establish intrinsic alignment between new and old prompts. (2) Online Alignment Preservation, which dynamically identifies the target old prompts and applies adaptive componential consistency constraints as new prompts evolve. Extensive experiments on DIL benchmarks demonstrate the effectiveness of our KA-Prompt. Our source code is available at https://github.com/zhoujiahuan1991/ICML2025-KA-Prompt.

[1]Wangxuan Institute of Computer Technology, Peking University, Beijing, China [2]School of Artificial Intelligence and Automation, Huazhong University of Science and Technology, Wuhan, China [3]Amazon.com, Inc, Bellevue, WA 98004, USA. Correspondence to: Jiahuan Zhou <jiahuanzhou@pku.edu.cn>.

*Proceedings of the $42^{nd}$ International Conference on Machine Learning*, Vancouver, Canada. PMLR 267, 2025. Copyright 2025 by the author(s).

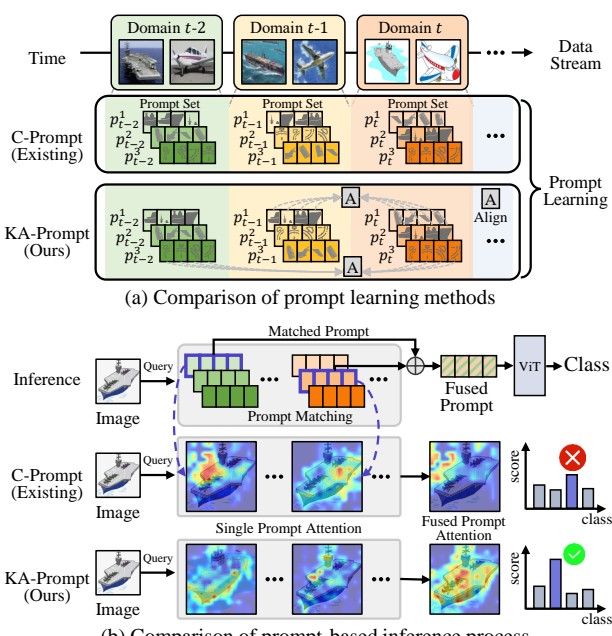

(a) Comparison of prompt learning methods

(b) Comparison of prompt-based inference process

Figure 1: (a) DIL aims to learn with a stream of data from different domains. State-of-the-art method C-Prompt learns domain-specific prompts independently, leading to component-aware (*e.g.*, object part orders) misalignment. To settle this, our KA-Prompt introduces cross-domain alignment constraints. (b) During inference, by fusing prompts from different domains, our KA-Prompt outperforms C-Prompt, benefiting from improved prompt learning capacity and enhanced cross-domain knowledge compatibility.

## 1. Introduction

Domain Incremental Learning (DIL) aims to adapt to continuous data streams with substantial domain shifts caused by variations in style, data quality degradation, and environmental changes (Song et al., 2024; Liu et al., 2024a; Gong et al., 2022; Shi et al., 2024a; Xu et al., 2025). The key difficulty of DIL lies in balancing the plasticity to learn new domains with the stability to retain and utilize historical knowledge, a trade-off known as the plasticity-stability or acquisition-forgetting dilemma (Li et al., 2024c).

Existing approaches to DIL primarily rely on data replay or regularization. Replay-based methods (Shi & Wang,

2024; Jeeveswaran et al., 2024) retain learned knowledge by storing historical samples, but they raise privacy concerns and require growing storage overhead. Regularization-based techniques (Wang et al., 2023b; Li & Hoiem, 2017) alleviate forgetting by constraining parameter updates, yet the strict constraints often hinder new knowledge acquisition.

Prompt-based methods have recently emerged as a promising alternative, where domain-specific prompts containing task knowledge can be stored in isolation to mitigate forgetting (Wang et al., 2022a). State-of-the-art approaches like C-Prompt (Liu et al., 2024a) underscored that fusing prompts across domains can improve model performance since cross-domain shared knowledge can be exploited to facilitate inference. However, as shown in Fig. 1 (a), since these methods learn prompts independently per domain, the prompts containing shared knowledge (*e.g.*, object parts knowledge) are randomly positioned within the componential level of the prompt. Therefore, the inter-domain prompts typically exhibit component-wise misalignment. This misalignment causes interference during cross-domain prompt fusion, limiting their ability to leverage shared knowledge.

To settle these problems, we propose KA-Prompt, a prompt-based framework that enforces component-wise knowledge alignment across domains through two key designs: (1) **Initial Componential Structure Configuring**: Instead of randomly initializing the components of new prompts, we actively mine a set of old prompts that contains shared knowledge with the new domain. By integrating their componential structures into the new prompts during initialization, an intrinsic cross-domain alignment is established. To achieve this, a *Reusable Knowledge Mining* mechanism is developed where a greedy prompt search algorithm is introduced to maximize the reusable knowledge within the selected prompt set. (2) **Online Alignment Preservation**: As the new prompts update, the componential structures within prompts typically drift and induce misalignment in prompts. To overcome this, we introduce a dynamic alignment mechanism to maintain the componential structure consistency between prompts. Specifically, an *Aligning-guided New Prompt Learning* scheme is exploited, where the target old prompts are dynamically identified to conduct adaptive alignment constraints as the new prompts evolve. Experimental results (*e.g.*, Fig. 1 (b)) verify that our method effectively enhances the knowledge compatibility between prompts, improving both the acquisition and inference capacity. Extensive experiments on DIL benchmarks show that our KA-Prompt outperforms the existing methods by large margins. To sum up, the contributions of the paper are as follows:

(1) Problem Identification: We reveal that component-wise misalignment in prompts limits their cross-domain knowledge integration and utilization capacity.

(2) Methodological Innovation: KA-Prompt introduces a component-aware alignment framework for prompt learning, proposing initial componential structure configuring to comprehensively mine reusable knowledge to obtain high-quality intrinsic alignment. Besides, online alignment preservation is developed to dynamically maintain the component-wise alignment as the prompts evolve.

(3) Extensive experiments conducted on four DIL benchmarks demonstrate the significant superiority of the proposed KA-Prompt over the state-of-the-art approaches.

## 2. Related Work

### 2.1. Incremental Learning

Incremental learning (IL) aims to continuously acquire new knowledge while retaining the old knowledge of historical tasks. Its primary challenge lies in balancing new knowledge acquisition with mitigating catastrophic forgetting (Li et al., 2024b; Chen et al., 2024; Kang & Choi, 2024). IL is commonly categorized into three scenarios. Task-incremental learning (TIL) assumes that task labels are available during inference (Oren & Wolf, 2021; Van de Ven et al., 2022). Class-incremental learning (CIL) sequentially learns new classes with task labels not provided during inference (Zhou et al., 2024; Li et al., 2024a), typically within a single domain. Domain-incremental learning (DIL) presents a more challenging setting where the class set remains unchanged, but each task introduces a new domain, and task labels are not available during inference (Mirza et al., 2022; Lamers et al., 2023). Larger domain shifts generally exacerbate the forgetting problem (Xu et al., 2024a; Cui et al., 2024). Existing DIL methods can be broadly categorized into replay-based, regularization-based, and prompt-based approaches.

Replay-based methods store a subset of historical exemplars and replay them during new task learning (Jeeveswaran et al., 2024; Shi & Wang, 2024; Xie et al., 2022). However, their reliance on raw data storage raises privacy risks and scalability concerns, especially in resource-constrained scenarios (Xu et al., 2024c) Regularization-based approaches impose constraints on model features or parameters to mitigate forgetting (Asadi et al., 2023; Bonato et al., 2024; Li et al., 2025), but such constraints can significantly hinder the acquisition of new domain knowledge due to the substantial representational gaps between domains (Xu et al., 2024b).

Recently, prompt-based learning has emerged as an effective solution for DIL (Wang et al., 2022a; 2023b; Liu et al., 2024a). By maintaining a pool of domain-specific prompts, these methods effectively preserve historical knowledge and mitigate catastrophic forgetting. Most prompt-based DIL approaches are implemented using pre-trained Vision Transformers (ViT)(Wang et al., 2021; Shi et al., 2024b; Zhang et al., 2025), and recent works (Feng et al., 2024; Wang

et al., 2025) have also explored multi-modal pre-trained models such as CLIP (Radford et al., 2021; Liu et al., 2025). This study focuses on ViT-based prompt to ensure a fair comparison with recent state-of-the-art approaches.

## 2.2. Prompt Composition in Incremental Learning

Recent studies (Smith et al., 2023; Liu et al., 2024a) have demonstrated that fusing prompts from different domains can enhance model robustness during inference. CODA-Prompt (Smith et al., 2023) learns soft weights for each prompt and combines all task prompts for inference. As domain knowledge varies significantly across prompts, fusing all prompts can lead to knowledge conflicts and suboptimal performance. To address this, C-Prompt (Liu et al., 2024a) selects and fuses only the most relevant prompts for each input sample, achieving state-of-the-art performance. However, since C-Prompt learns prompts independently for each domain, the misalignment of knowledge across prompts hinders their effective integration, thereby limiting performance gains from prompt composition. To overcome these limitations, this paper focuses on enhancing the utilization of historical knowledge and improving cross-domain prompt fusion compatibility to optimize inference performance.

## 3. The Proposed Method

This section first presents the preliminaries in Sec. 3.1, including the problem formulation of DIL and an overview of the C-Prompt baseline. Then, a motivation study is provided in Sec. 3.2 to analyze the phenomenon of prompt misalignment. Finally, the proposed KA-Prompt framework is detailed in Sec. 3.3.

### 3.1. Preliminary

**Problem Formulation:** In DIL, the model is expected to train with a stream of domains to become a universal expert for all the domains. Formally, a stream of $T$ datasets $\mathcal{D} = \{D_t\}_{t=1}^{T}$ is given step by step during training, where $D_t = \{(\boldsymbol{x}_i, y_i)\}_{i=1}^{N_t}$ contains $N_t$ pairs of image $\boldsymbol{x}_i$ and corresponding label $y_i$. When $D_t$ is given, the previous $t-1$ datasets are inaccessible (Liu et al., 2024a). Each domain is collected with a test dataset $D_t^{te}$ for model evaluation.

**Compositional Prompting (C-Prompt):** A pre-trained Vision Transformer (ViT) model $f_\theta$ is employed as the backbone, with its parameters $\theta$ kept frozen throughout training. To adapt to each domain, a learnable prompt set $\mathcal{P}_t$ is introduced, defined as:

$$\mathcal{P}_t = \{(\boldsymbol{p}_t^1, \boldsymbol{k}_t^1), (\boldsymbol{p}_t^2, \boldsymbol{k}_t^2), \ldots, (\boldsymbol{p}_t^{N_p}, \boldsymbol{k}_t^{N_p})\}, \quad (1)$$

where each tuple $(\boldsymbol{p}_t^i, \boldsymbol{k}_t^i)$ consists of a prompt embedding $\boldsymbol{p}_t^i \in \mathbb{R}^{L_p \times D}$ and a key $\boldsymbol{k}_t^i \in \mathbb{R}^D$. Here, $L_p$ denotes the prompt length, and $D$ represents the embedding dimension.

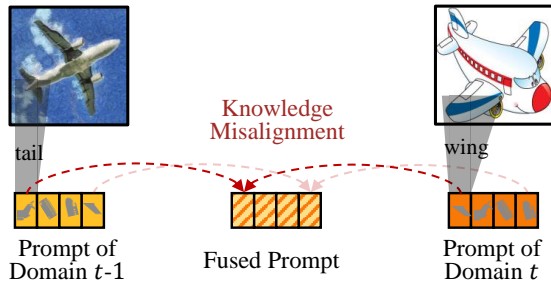

(a) Knowledge misalignment in C-Prompt (Liu et al., 2024a)

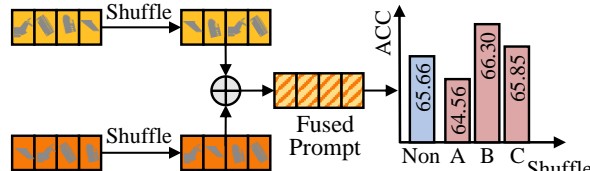

(b) Evaluation by shuffling prompt components before fusion

Figure 2: (a) Within a prompt, different components typically encode distinct types of knowledge. In C-Prompt, independently learned prompts exhibit misalignment in componential knowledge, leading to the fusion of irrelevant knowledge during inference. (b) By shuffling the componential positions of different domains before fusion, some orders with better knowledge alignment can be generated.

The key $\boldsymbol{k}_t^i$ is learnable and used for prompt matching.

Given an input image $\boldsymbol{x} \in \mathbb{R}^{H \times W \times C}$, where $H$, $W$, and $C$ denote the height, width, and number of channels, respectively, it is first tokenized (Ronen et al., 2023) into a sequence representation $\boldsymbol{h_x} \in \mathbb{R}^{L_h \times D}$, where $L_h$ is the sequence length. Then, the following process is adopted to obtain model prediction according to $\mathcal{P}_t$:

Firstly, a query function $q(\cdot)$, using $f_\theta$ as feature extractor, is exploited to encode the image into a $D$-dimensional query vector, *i.e.*, $q(\boldsymbol{x}) \in \mathbb{R}^D$. Next, the similarity between the input sample and a given prompt key $\boldsymbol{k}_t^i$ is computed as:

$$\mathcal{S}(q(\boldsymbol{x}), \boldsymbol{k}_t^i) = \left(1 + cos(q(\boldsymbol{x}), \boldsymbol{k}_t^i)\right)/2, \quad (2)$$

where $cos(\cdot)$ denotes the cosine similarity function. $\mathcal{S}(q(\boldsymbol{x}), \boldsymbol{k}_t^i) \in [0, 1]$ and a higher $\mathcal{S}(q(\boldsymbol{x}), \boldsymbol{k}_t^i)$ value indicates stronger relevance between $\boldsymbol{x}$ and $\boldsymbol{p}_t^i$.

The top-K relevant prompts, represented as $\mathcal{K}_{\boldsymbol{x}} = \{\dot{\boldsymbol{p}}_t^1, \dot{\boldsymbol{p}}_t^2, \ldots, \dot{\boldsymbol{p}}_t^K\}$, are selected and then fused into a compositional prompt $\overline{\boldsymbol{p}}_{\boldsymbol{x}}^t$ via a linear combination $g_c(\cdot)$ process:

$$\overline{\boldsymbol{p}}_{\boldsymbol{x}}^t = g_c(\dot{\boldsymbol{p}}_t^1, \dot{\boldsymbol{p}}_t^2, \ldots, \dot{\boldsymbol{p}}_t^K) = \frac{1}{K}\sum_{i=1}^{K} \dot{\boldsymbol{p}}_t^i \quad (3)$$

Then, $\overline{\boldsymbol{p}}_{\boldsymbol{x}}^t$ is concatenated with a class token $[cls] \in \mathbb{R}^{1 \times D}$

and $\boldsymbol{h_x}$, forming the final input representation $\boldsymbol{h_x^*} = \left[[cls]; \boldsymbol{h_x}; \overline{\boldsymbol{p}}_{\boldsymbol{x}}^t\right] \in \mathbb{R}^{(1+L_h+L_p) \times D}$. This representation is subsequently processed by the self-attention layers. A classification head is ultimately used to get the class prediction.

### 3.2. Motivational Study

Prompts are a few mounts of additional learnable parameters that guide the model to focus on the discriminative features (Pu et al., 2024; Liu et al., 2024b; Yao et al., 2025). As illustrated in Fig. 2 (a), within a prompt, $\boldsymbol{p}_t^i \in \mathbb{R}^{L_p \times d}$, its $L_p$ components can capture distinct discriminative knowledge, such as object parts (head, wing, tail, ...). Since C-Prompt learns each prompt set $\mathcal{P}_t$ independently during training, the corresponding prompt components across different domains become misaligned. Consequently, during prompt fusion, unrelated knowledge components are merged, leading to knowledge conflicts and suboptimal model performance.

To validate this issue, we conduct an experiment on C-Prompt by randomly shuffling prompt components across different domains before fusion during evaluation. The results, shown in Fig 2 (b), are obtained by training on the DomainNet benchmark and evaluating accuracy (ACC), where the last domain is adopted for analysis. "Non" represents the original evaluation results, while "A," "B," and "C" correspond to three rounds of random shuffling. The observed fluctuations in accuracy suggest that the original knowledge component order learned by C-Prompt is far from optimal. Therefore, one of our objectives is to improve cross-domain knowledge alignment to boost model inference.

Furthermore, since the domain-specific knowledge is learned independently, the generalizable knowledge acquired from previous domains remains underutilized when C-Prompt learns new domains. This limitation hinders C-Prompt's ability to adapt to new domains. Hence, another key objective of this work is to enhance the utilization of historical knowledge to improve new domain learning.

### 3.3. KA-Prompt Approach

The detailed design of our KA-Prompt is illustrated in Fig. 3. It consists of a Reusable Knowledge Mining mechanism ($\boldsymbol{\Psi}_{\mathcal{M}}$) and an Aligning-Guided New Prompt Learning scheme ($\boldsymbol{\Psi}_{\mathcal{L}}$), which serve to extract generalizable knowledge from previous domains to facilitate new prompt learning and to ensure cross-domain prompt alignment, respectively.

### Reusable Knowledge Ming

A straightforward approach to fully utilize historical knowledge and ensure cross-domain knowledge alignment is to directly transfer historical prompts to new prompts. However, since the total size of the historical prompt set, $|\mathcal{P}_1 \cup \mathcal{P}_2 \cup \cdots \cup \mathcal{P}_{t-1}|$, grows proportionally to $t-1$ times

the size of a single prompt set $|\mathcal{P}_t| = N_{\boldsymbol{p}}$, an appropriate selection of $N_{\boldsymbol{p}}$ historical prompts is required. While certain approaches initialize new prompts using those from the preceding stage (Wang et al., 2023a), they are limited in their ability to capture the shared knowledge between old and new domains, as the most relevant prior knowledge may be dispersed across multiple domains. For example, in a domain order of *Real→Quickdraw→Clipart*, some *Clipart* images may exhibit similarities to the *Quickdraw* style, whereas others may be more closely related to the *Real* style. Hence, two key principles guide the selection of reusable prompts:

· *The selected prompts should contain knowledge that is highly relevant to the new domain.*

· *The selected prompts should collectively cover as much new domain knowledge as possible.*

A direct method to fulfill these principles is exhaustive grid search, which requires $\mathcal{C}_{(t-1) \times L_s}^{L_s} = O(n^k)$ iterations. Consequently, this approach is NP-hard and computationally prohibitive. To address this, a Reusable Knowledge Mining module is proposed, employing a greedy search strategy (Flamich, 2024; Yang et al., 2024).

**Prompt-Data Relation Extraction ($\boldsymbol{f}_R$):** Given a new domain dataset $D_t$, the relationship between historical prompts and new domain samples is first assessed. Specifically, the pre-trained ViT $f_\theta$ is used to extract image features from $D_t$, forming a feature pool $\mathbf{F}_t \in \mathbb{R}^{N_t \times D}$. Meanwhile, the keys of all historical prompts are arranged into a matrix $\mathbf{K}_{t-1} = [\boldsymbol{k}_1^1; \boldsymbol{k}_1^2; \ldots; \boldsymbol{k}_1^{N_p}; \boldsymbol{k}_2^1; \ldots; \boldsymbol{k}_{t-1}^{N_p}] \in \mathbb{R}^{[(t-1) \times N_p] \times D}$. An base relation matrix $\mathbf{S}^0 \in \mathbb{R}^{[(t-1) \times N_p] \times N_t}$ is then computed as

$$\mathbf{S}^0 = [f_n(\mathbf{K}_{t-1}) f_n^\top(\mathbf{F}_t) + 1]/2, \tag{4}$$

where $f_n(\cdot)$ denotes L2 normalization applied row-wise to the matrix. Each element $\mathbf{S}_{i,j}^0 \in [0,1]$ represents the correlation score between a historical prompt and a new domain sample.

**Greedy Prompt Search ($\boldsymbol{f}_G$):** A reusable prompt memory $\mathcal{M}_t$ is initialized as an empty set, which progressively includes the historical prompts most relevant to the new domain. At each step, the prompt containing the highest amount of unique new domain-relevant knowledge (absent from $\mathcal{M}_t$) is identified and added to $\mathcal{M}_t$. This process is repeated until $\mathcal{M}_t = N_{\boldsymbol{p}}$.

Specifically, at each search step, we construct a memory relation matrix $\mathbf{S}^{\mathcal{M}} \in \mathbb{R}^{|\mathcal{M}_t| \times N_t}$ by selecting rows from the base matrix $\mathbf{S}^0$ according to the indices of filtered reusable historical prompts. A sample effect vector $\boldsymbol{v} \in \mathbb{R}^{N_t}$ is then computed by selecting the maximum value from each column of $\mathbf{S}^{\mathcal{M}}$, representing the degree to which each new sample has already been assigned relevant knowledge. This

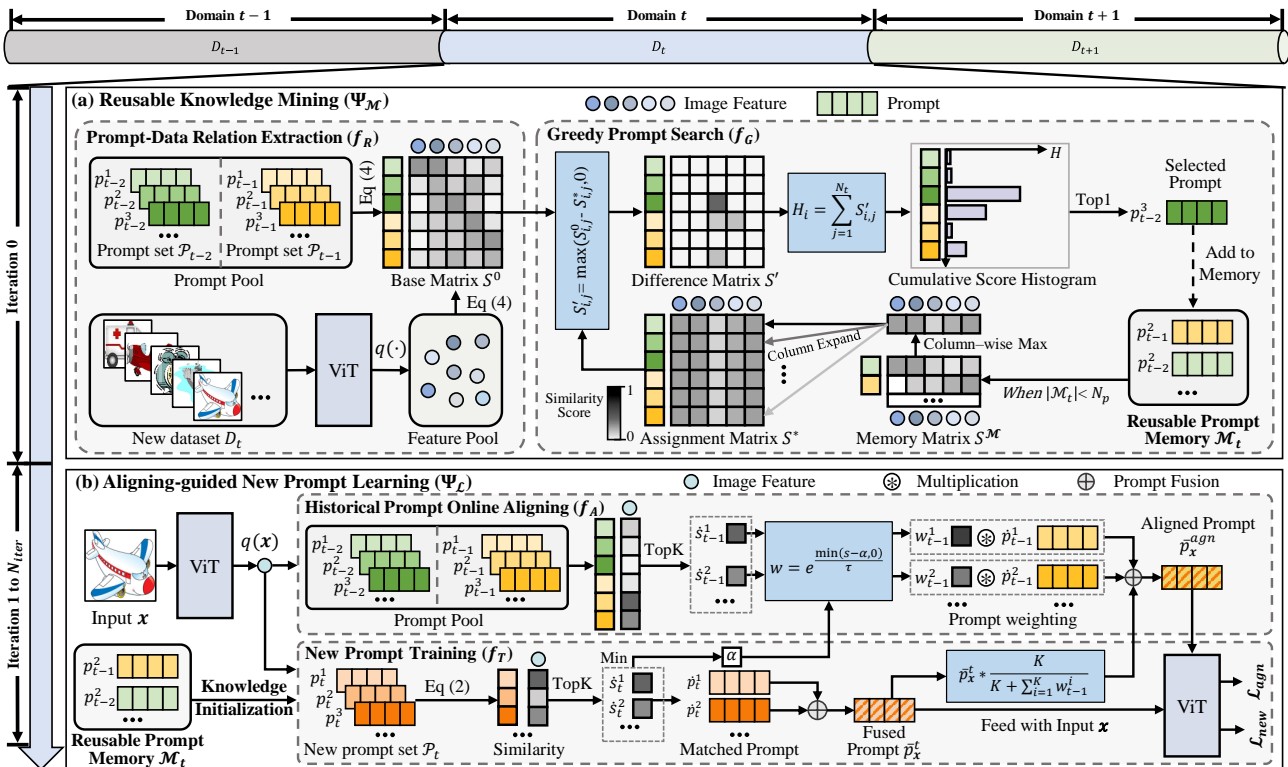

Figure 3: The illustration of our KA-Prompt method. When the new domain data $D_t$ is given, the Reusable Knowledge Mining mechanism constructs a reusable prompt memory that contains the shared knowledge between the old and new domains. The reusable prompt memory is then utilized to initiate new prompts. Next, an Aligning-guided New Prompt Learning scheme is conducted for $N_{iter}$ iterations to learn the knowledge of the new domain, where the new prompt training and historical prompt online aligning ensure new knowledge acquisition and cross-domain aligning, respectively.

vector is then expanded into $\mathbf{S}^* \in \mathbb{R}^{[(t-1) \times N_P] \times N_t}$ which is denoted assignment matrix . Then, a difference matrix

$$\mathbf{S}'_{i,j} = \max\{\mathbf{S}^0_{i,j} - \mathbf{S}^*_{i,j}, 0\}. \quad (5)$$

is computed to quantify the additional unique knowledge that each historical prompt contributes. The overall unique relevant knowledge of each prompt is then estimated via a cumulative score histogram

$$H_i = \sum_{j=1}^{N_t} \mathbf{S}'_{i,j}. \quad (6)$$

The prompt corresponding to the top-1 value in $H$ is added to $\mathcal{M}_t$. The above process is repeated until $\mathcal{M}_t$ is complete.

**Aligning-guided New Prompt Learning**

Once $\mathcal{M}_t$ is obtained, we initialize new domain prompts using $\mathcal{M}_t$ to achieve a strict componental structure alignment between new and old prompts. However, despite the initial alignment, componental structure drift can occur as new prompts update during training. To mitigate this, we introduce the following learning and alignment constraints.

**New Prompt Training ($f_T$):** Given an input image $x$, the pre-trained ViT-based query function $q(x)$ is employed to extract an image query. The prompt matching and fusing process of C-Prompt, introduced in Sec. 3.1, is then applied to obtain the fused prompt $\overline{p}_x^t$ is obtained, which is fed to the pre-trained ViT with the input image $x$. A classification loss is used to facilitate prompt learning:

$$\mathcal{L}_{new} = \mathrm{CE}(f_\theta(x, \overline{p}_x^t), y), \quad (7)$$

where CE represents the cross-entropy loss, and $y$ is the label of $x$. Besides, as shown in Fig. 3 (b), the Top-K matched similarity score set $\mathcal{S}_t = \{\dot{s}_t^1, \dot{s}_t^2, \ldots, \dot{s}_t^K\}$ are extracted in $f_T$ module, and the minimal matched similarity $\alpha = \min\{S_t\}$ is retained for subsequent process.

**Historical Prompt Online Aligning ($f_A$):** Given the historical prompt pool $\tilde{\mathcal{P}}_{t-1} = \mathcal{P}_1 \cup \mathcal{P}_2 \cup \cdots \cup \mathcal{P}_{t-1}$, the prompt matching process of C-Prompt, as introduced in Sec. 3.1, is applied. The resulting matched scores and prompts are denoted as $\{\dot{s}_{t-1}^1, \dot{s}_{t-1}^2, \ldots, \dot{s}_{t-1}^K\}$ and $\{\dot{p}_{t-1}^1, \dot{p}_{t-1}^2, \ldots, \dot{p}_{t-1}^K\}$ respectively. Each matched historical prompt $\dot{p}_{t-1}^i$ is assigned a prompt weight $w_{t-1}^i$, com-

Table 1: The Avg-ACC performance comparison across four DIL benchmarks.

| Methods | Publication | DomainNet | ImageNet-R | ImageNet-C | ImageNet-Mix | Average |
|---|---|---|---|---|---|---|
| EWC | NAS 2017 | $47.10_{\pm0.58}$ | $51.83_{\pm0.33}$ | $65.20_{\pm1.23}$ | $57.82_{\pm0.90}$ | $55.49_{\pm0.42}$ |
| LwF | T-PAMI 2017 | $54.85_{\pm0.06}$ | $57.11_{\pm0.95}$ | $69.01_{\pm1.21}$ | $65.27_{\pm0.93}$ | $61.56_{\pm0.45}$ |
| L2P | CVPR 2022 | $53.74_{\pm0.04}$ | $56.55_{\pm0.33}$ | $77.86_{\pm0.44}$ | $64.20_{\pm0.34}$ | $63.09_{\pm0.16}$ |
| S-Prompts | NeurIPS 2022 | $44.08_{\pm0.05}$ | $27.23_{\pm0.16}$ | $60.25_{\pm0.28}$ | $24.25_{\pm0.19}$ | $38.95_{\pm0.09}$ |
| DualPrompt | ECCV 2022 | $55.18_{\pm0.02}$ | $59.47_{\pm1.00}$ | $78.52_{\pm0.30}$ | $63.57_{\pm0.17}$ | $64.19_{\pm0.26}$ |
| ESN | AAAI 2023 | $45.74_{\pm0.25}$ | $16.39_{\pm0.23}$ | $68.34_{\pm0.39}$ | $14.99_{\pm0.53}$ | $36.37_{\pm0.19}$ |
| CODA-Prompt | CVPR 2023 | $55.81_{\pm0.03}$ | $55.21_{\pm0.33}$ | $78.25_{\pm0.16}$ | $64.92_{\pm0.04}$ | $63.55_{\pm0.09}$ |
| InfLoRA | CVPR 2024 | $48.76_{\pm0.30}$ | $41.20_{\pm1.65}$ | $53.12_{\pm0.16}$ | $35.38_{\pm0.27}$ | $44.62_{\pm0.43}$ |
| Cprompt | CVPR 2024 | $52.88_{\pm0.12}$ | $59.48_{\pm1.09}$ | $70.08_{\pm0.07}$ | $63.64_{\pm0.74}$ | $61.52_{\pm0.33}$ |
| C-Prompt | IJCV 2024 | $\underline{58.66}_{\pm0.05}$ | $\underline{62.43}_{\pm0.49}$ | $\underline{79.84}_{\pm0.38}$ | $\underline{65.35}_{\pm0.52}$ | $\underline{66.57}_{\pm0.20}$ |
| KA-Prompt | This Paper | $\mathbf{62.91}_{\pm0.14}$ | $\mathbf{66.51}_{\pm0.36}$ | $\mathbf{85.43}_{\pm0.56}$ | $\mathbf{70.35}_{\pm0.32}$ | $\mathbf{71.30}_{\pm0.19}$ |

puted as follows:

$$w_{t-1}^i = e^{\min\{\dot{s}_{t-1}^i - \alpha, 0\}/\tau}, \tag{8}$$

where $\tau$ is a hyperparameter used to scale the weight. If an old prompt exhibits greater similarity to the query than the matched new prompt, its weight $w_{t-1}^i$ is set to 1. Conversely, when an old prompt is less similar to the query than the matched new prompt, its weight $w_{t-1}^i$ decreases proportionally to the similarity reduction.

To maintain the componental structure alignment of cross-domain prompts, the prompts are fused as follows:

$$\overline{\boldsymbol{p}}_{\boldsymbol{x}}^{agn} = \frac{1}{K + \sum_{i=1}^{K} w_{t-1}^i} \sum_{i=1}^{K} (w_{t-1}^i \dot{\boldsymbol{p}}_{t-1}^i) + \frac{K}{K + \sum_{i=1}^{K} w_{t-1}^i} \overline{\boldsymbol{p}}_{\boldsymbol{x}}^t, \tag{9}$$

where the term $\frac{1}{K + \sum_{i=1}^{K} w_{t-1}^i}$ is employed to normalize the overall weights.

Finally, $\overline{\boldsymbol{p}}_{\boldsymbol{x}}^{agn}$ is fed to the pre-trained ViT with the input image $\boldsymbol{x}$. A cross-entropy-based prompt knowledge alignment loss is introduced to ensure the componental fusion compatibility between new and old prompts:

$$\mathcal{L}_{agn} = \text{CE}(f_\theta(\boldsymbol{x}, \overline{\boldsymbol{p}}_{\boldsymbol{x}}^{agn}), y). \tag{10}$$

**Training and Inference**

During training, once $D_t$ is obtained, $\boldsymbol{\Psi}_{\mathcal{M}}$ (as shown in Fig. 3 (a)) is executed for one time. Then, $\boldsymbol{\Psi}_{\mathcal{L}}$ is applied $N_{iter}$ iterations under the supervision of the overall loss:

$$\mathcal{L} = \mathcal{L}_{new} + \lambda \mathcal{L}_{agn}, \tag{11}$$

where $\lambda$ is a hyperparameter to balance the new knowledge learning and cross-domain prompt alignment.

During inference, the prompt sets of all encountered domains are gathered, i.e., $\mathcal{P} = \mathcal{P}_1 \cup \mathcal{P}_2 \cup \cdots \cup \mathcal{P}_T$. The top-K matched prompts are then selected from $\mathcal{P}$, and the same pipeline as in new prompt training is followed to generate classification results.

## 4. Experiments

### 4.1. Experimental Settings

**Datasets:** Our experiments are conducted on four multi-domain benchmarks including DomainNet, ImageNet-R, ImageNet-C, and ImageNet-Mix. Within each benchmark, the domains are sorted by decreasing image counts to simulate a challenging DIL scenario, following C-Prompt (Liu et al., 2024a) and CaSSLe (Fini et al., 2022).

DomainNet (Peng et al., 2019) contains 345 classes and 586,575 images which are collected from six different style domains, i.e., *Real, Quickdraw, Sketch, Painting, Infograph,* and *Clipart*. The training data across classes and domains are imbalanced, making this dataset more challenging.

ImageNet-R (Hendrycks et al., 2021) contains 30,000 images of 200 categories. All images are split into 15 different style domains. The images in each domain are divided into training and testing sets with a 7:3 ratio.

ImageNet-C (Hendrycks & Dietterich, 2018) contains 1000 categories covering 15 quality corruptions and environmental changes. Following (Liu et al., 2024a), 200 categories identical to ImageNet-R in ImageNet-C are used to form a DIL benchmark, where each category contains 7,000 images for training and 3,000 images for testing.

ImageNet-Mix (Liu et al., 2024a) is built by fusing ImageNet-C and ImageNet-R, which comprises a total of 30 domains, incorporating various image styles, qualities, and environmental variations.

**Evaluation Metrics:** Following previous works (Liu et al., 2024a; Wang et al., 2022a), given a DIL dataset with $T$ do-

Table 2: The per-domain performance comparison on DomainNet. The domain order during training is *Real→Quickdraw→ Painting→Sketch→Infograph→Clipart*.

| Method | Publication | *Real* | *Quickdraw* | *Painting* | *Sketch* | *Infograph* | *Clipart* | **Avg-ACC** |
|---|---|---|---|---|---|---|---|---|
| EWC | *NAS 2017* | 60.57±0.56 | 25.43±1.5 | 44.55±1.16 | 50.06±0.18 | 25.69±0.56 | 76.29±0.37 | 47.10±0.58 |
| LwF | *T-PAMI 2017* | 68.52±0.05 | 33.10±0.45 | 54.80±0.32 | 58.54±0.31 | 35.45±0.60 | 78.67±0.20 | 54.85±0.06 |
| L2P | *CVPR 2022* | 77.96±0.12 | 19.09±0.11 | 59.75±0.17 | 56.38±0.04 | 30.52±0.13 | 78.74±0.06 | 53.74±0.43 |
| S-Prompts | *NeurIPS 2022* | 65.54±0.08 | 8.42±0.38 | 47.24±0.12 | 43.96±0.03 | 20.99±0.02 | 78.35±0.09 | 44.08±0.05 |
| DualPrompt | *ECCV 2022* | 78.11±0.01 | 24.36±0.12 | 60.67±0.11 | 57.85±0.14 | 30.88±0.07 | 79.23±0.11 | 55.18±0.02 |
| ESN | *AAAI 2023* | 65.95±0.11 | 9.87±0.39 | 48.29±0.49 | 48.93±0.56 | 22.17±0.27 | 79.24±0.13 | 45.74±0.25 |
| CODA-Prompt | *CVPR 2023* | 78.37±0.03 | 23.89±0.03 | 60.33±0.03 | 59.98±0.07 | 31.82±0.01 | **80.46**±0.08 | 55.81±0.03 |
| InfLoRA | *CVPR 2024* | 66.36±0.32 | 18.54±0.64 | 49.17±0.25 | 52.47±0.27 | 26.18±0.25 | 79.83±0.24 | 48.76±0.30 |
| Cprompt | *CVPR 2024* | **83.47**±0.08 | 26.24±0.46 | 67.30±0.26 | 45.78±0.29 | 28.90±0.16 | 65.58±0.19 | 52.88±0.12 |
| C-Prompt | *IJCV 2024* | 83.34±0.11 | 49.17±0.30 | 64.55±0.23 | 57.20±0.22 | 32.29±0.10 | 65.44±0.06 | 58.66±0.05 |
| KA-Prompt | *This Paper* | 82.30±0.07 | **52.12**±0.26 | **68.42**±0.15 | **62.43**±0.23 | **38.25**±0.16 | 72.97±0.38 | **62.91**±0.14 |

mains, the average accuracy (**Avg-ACC**) is used to evaluate the model. It is calculated as:

$$A_T = \frac{1}{T} \sum_{i=1}^{T} a_{T,i}, \tag{12}$$

where $a_{T,i}$ represents the classification accuracy on the $i$-th domain testing with the model trained on $T$-th domain ($\mathrm{M}_T$), and $A_T$ represents the overall performance of $\mathrm{M}_T$ across different domains.

Additionally, the **Average** performance across 4 DIL benchmarks is reported to assess the overall DIL capacity of different models in diverse scenarios.

**Compared Methods:** We compare our KA-Prompt with the regularization-based incremental learning methods LwF (Li & Hoiem, 2017), EWC (Kirkpatrick et al., 2017), ESN (Wang et al., 2023b) and prompt-based method L2P (Wang et al., 2022c), S-Prompts (Wang et al., 2022a), DualPrompt (Wang et al., 2022b), CODA-Prompt (Smith et al., 2023), CPrompt (Gao et al., 2024) and C-Prompt (Liu et al., 2024a). Besides, we also compare with the parameter-efficient tuning method InfLoRA (Liang & Li, 2024). All experiments are conducted under the official codes with ViT (ViT-B/16) pre-trained on ImageNet-21k as the backbone.

**Implementation Details:** We follow the prompt and classifier configuration of C-Prompt (Liu et al., 2024a), *e.g.*, prompt length $L_{\boldsymbol{p}}$ and prompt number $N_{\boldsymbol{p}}$ of each domain, the shared classifier across all domains. The Adam optimizer ($\beta_1 = 0.9$, $\beta_2 = 0.999$) is adopted to train the model. The default batch size and learning rate for all benchmarks are set to 128 and 0.005 respectively, except for DomainNet where the learning rate is set to 0.0006. The default training epochs are set to 5 except for DomainNet (10 epochs). The training images are resized to 224×224. The hypermeters $\tau$ and $\lambda$ are set to 0.01 and 0.1 by default, respectively. All experiments are conducted on a single Nvidia 4090 GPU.

## 4.2. Comparison with State-of-the-Art

The comparison results across four DIL datasets, along with the average performance, are presented in Tab. 1. Specifically, our KA-Prompt outperforms the state-of-the-art C-Prompt, achieving improvements of **4.25%/4.08%/5.59%/5.00%** on DomainNet, ImageNet-R, ImageNet-C, and ImageNet-Mix, respectively. Overall, KA-Prompt surpasses C-Prompt with an average improvement of **4.73%** across the four datasets. These results underscore the robustness of KA-Prompt to variations in style and data quality, attributed to its enhanced utilization of historical knowledge during training and improved cross-domain knowledge compatibility that boosts discriminative feature utilization during inference.

Additionally, Tab. 2 reports the detailed per-domain performance on DomainNet, highlighting KA-Prompt's superior performance on intermediate domains such as Quickdraw, Painting, Sketch, and Infograph. We also observe that C-Prompt and CODA-Prompt outperform KA-Prompt on the first and last domains, respectively. This discrepancy can be attributed to C-Prompt's emphasis on domain-specific knowledge isolation during training, which reduces forgetting but limits the model's acquisition capacity of new domains. On the other hand, other prompt-based methods, *e.g.*, CODA-Prompt, L2P, S-Prompts, and DualPrompt, prioritize new knowledge acquisition but suffer from catastrophic forgetting caused by prompt-classifier misalignment. In contrast, our reusable prompt mining and initialization design effectively utilize generalizable knowledge of historical domains to enhance new domain learning. Besides, the online aligning design consolidates the componential knowledge alignment as new prompts evolve, improving the robustness of cross-domain knowledge utilization during inference.

To further analyze the model learning process, we visualize the Avg-ACC performance across seen domains through-

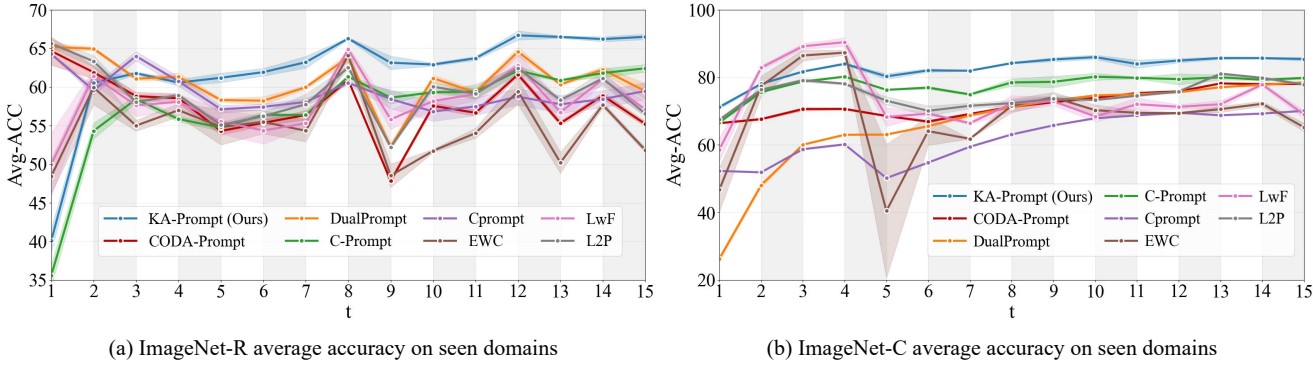

(a) ImageNet-R average accuracy on seen domains

(b) ImageNet-C average accuracy on seen domains

Figure 4: The seen domain performance tendency along domain incremental learning process.

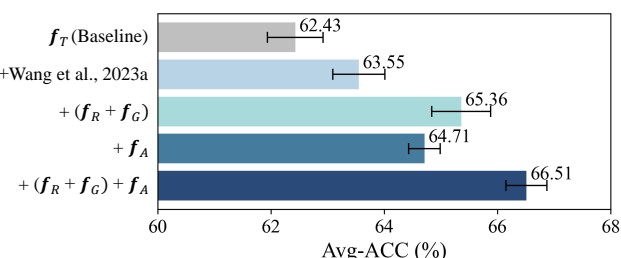

Figure 5: Ablation on the model components.

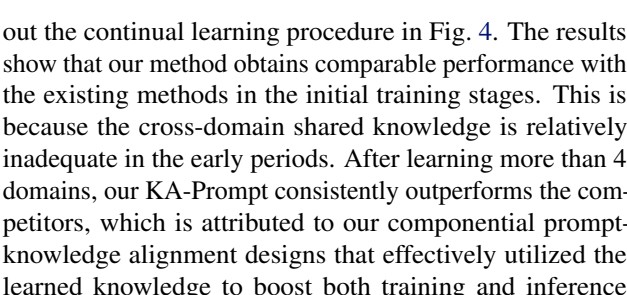

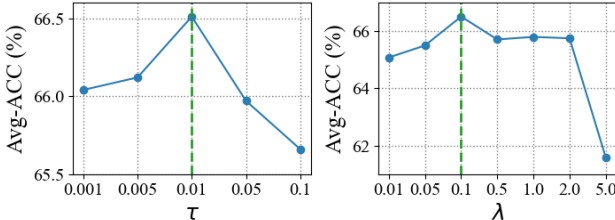

Figure 6: Ablation studies on the hyper-parameters under ImageNet-R dataset.

out the continual learning procedure in Fig. 4. The results show that our method obtains comparable performance with the existing methods in the initial training stages. This is because the cross-domain shared knowledge is relatively inadequate in the early periods. After learning more than 4 domains, our KA-Prompt consistently outperforms the competitors, which is attributed to our componential prompt-knowledge alignment designs that effectively utilized the learned knowledge to boost both training and inference along the continual learning process.

### 4.3. Ablation Studies

**Analysis on the proposed modules**. In Fig. 5, we conduct ablation studies on the modules of our KA-Prompt. When using the $f_T$ module alone, our method degrades as C-Prompt (Liu et al., 2024a) baseline. $f_R$ and $f_G$ do not influence the model when used alone, and they form $\Psi_{\mathcal{M}}$ when used together. To further verify the historical knowledge utilization capacity of $\Psi_{\mathcal{M}}$, we additionally compare with (Wang et al., 2023a) that provides a simple initial prompt initialization strategy, *i.e.*, using the prompt of the previous stage $\mathcal{P}_{t-1}$ to initialize $\mathcal{P}_t$.

The experimental results show that $f_T + (f_R + f_G)$ achieves **1.81%** improvement compared to $f_T +$(Wang et al., 2023a), demonstrating effectively mine the generalizable knowl-

edge from old domains. Besides, $f_T + f_A$, *i.e.*, $\Psi_{\mathcal{L}}$, obtains **2.28%** improvement compared to the baseline, verifying the effectiveness of online alignment design. Finally, when all our modules are used together, the model performance is further improved since the training-stage historical knowledge utilization and the inference stage knowledge compatibility is complementary to each other.

**Analysis on the hyper-parameters** In Fig. 6, we evaluate the performance of KA-Prompt under different values of the hyperparameters $\tau$ and $\lambda$. The parameter $\tau$ controls the weights of old prompts that are distant from new samples during prompt fusion, where a larger $\tau$ increases their impact. Meanwhile, $\lambda$ serves as the weight for the alignment loss, balancing new knowledge learning and componential alignment to the historical prompts. Based on empirical analysis, the optimal hyperparameter values are set to $\tau = 0.01$ and $\lambda = 0.1$ as the default configuration.

**Ablation on prompt shuffle.** To demonstrate that our componential prompt-knowledge alignment reduces knowledge conflicts, we conduct an ablation study by shuffling prompt components under varying conditions. As shown in Fig. 7 (a), C-Prompt (Liu et al., 2024a) experiences performance improvement in Shuffle-B/C/D/E due to its randomly learned componential structure, making the original componential structure tend to be suboptimal. In contrast, KA-Prompt consistently exhibits degradation when prompt

components are perturbed. KA-Prompt's sensitivity to component structures confirms its success in learning an intrinsically aligned structure.

Besides, although random shuffling introduces misalignment noise that disrupts KA-Prompt's carefully organized components, the worst performance of KA-Prompt (Fig. 7 (b) Shuffle-E) is **6.4%** superior to the best condition of C-Prompt (Fig. 7 (a) Shuffle-B). This is attributed to our greedy prompt search algorithm effectively collecting the generalizable knowledge from all historical domains, significantly improving the new domain adaptation capacity of the model.

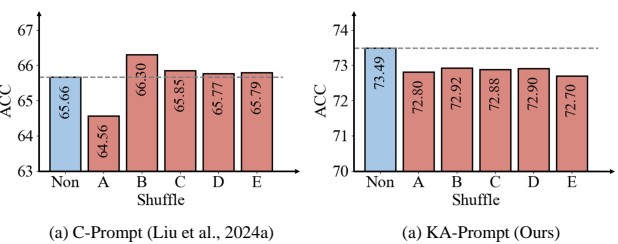

(a) C-Prompt (Liu et al., 2024a)     (a) KA-Prompt (Ours)

Figure 7: Performance comparison on the last domain of DomainNet under different componential position shuffling conditions.

## 5. Conclusion

In this paper, we investigate the practical and challenging domain incremental learning problem and propose a novel method KA-Prompt. Based on the observation that component-wise misalignment between domain-specific prompts leads to conflicting knowledge integration and degraded predictions, we introduce a component-wise knowledge alignment paradigm with two complementary designs: (1) Initial Componential Structure Configuring exploits a novel greedy prompt search algorithm to comprehensively mine the historical prompts containing reusable knowledge, which are utilized to provide an effective knowledge transfer and intrinsic alignment. (2) Online Alignment Preservation dynamically identifies the target old prompts and applies adaptive componential consistency constraints along the new prompt learning procedure. Extensive experimental results underscore our component-wise alignment paradigm effectively improves the acquisition and inference capacity simultaneously.

## Acknowledgements

This work was supported by the National Natural Science Foundation of China (62376011) and the National Key R&D Program of China (2024YFA1410000).

## Impact Statement

This paper presents work whose goal is to advance the investigation of domain incremental learning in Computer Vision. There are many potential societal consequences of our work, including the continuously evolving AI agents, the multi-domain downstream stak adaption of large vision models, and the lifelong environment adaption of autonomous driving systems.

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

## A. Algorithm.

The overall process of our key phases $\Psi_{\mathcal{M}}$ and $\Psi_{\mathcal{L}}$ are shown in Alg. 1 and Alg. 2, respectively.

---

**Algorithm 1** Reusable Knowledge Mining ($\Psi_{\mathcal{M}}$)

---

**Input:** Data $D_t = \{(\boldsymbol{x}_i, y_i)\}_{i=1}^{N_t}$, Prompt pool $\mathcal{P} = \mathcal{P}_1 \cup \mathcal{P}_2 \cup \cdots \cup \mathcal{P}_{t-1}$
**Output:** Reusable prompt memory $\mathcal{M}_t$

*# Prompt-Data Relation Extraction module ($\boldsymbol{f}_R$)*
Extract image features $\mathbf{F}_t \in \mathbb{R}^{N_t \times D}$ from $D_t$ via pre-trained ViT $f_\theta$;
Gather the keys of prompts in $\mathcal{P}$: $\mathbf{K}_{t-1} = [\boldsymbol{k}_1^1; \boldsymbol{k}_1^2; \ldots; \boldsymbol{k}_1^{N_p}; \boldsymbol{k}_2^1; \ldots; \boldsymbol{k}_{t-1}^{N_p}]$;
Base relation matrix $\mathbf{S}^0 = [f_n(\mathbf{K}_{t-1}) f_n^\top(\mathbf{F}_t) + 1]/2$, Eq. 4;

*# Greedy Prompt Search module ($\boldsymbol{f}_G$)*
Initialize $\mathcal{M}_t = \emptyset$;
**for** $|\mathcal{M}_t| < N_p$ **do**
    Extract relation matrix $\mathbf{S}^{\mathcal{M}} \in \mathbb{R}^{|\mathcal{M}_t| \times N_t}$ according to $\mathcal{M}_t$ and $\mathbf{S}^0$;
    Obtain sample effect vector $\boldsymbol{v} \in \mathbb{R}^{N_t}$ by keeping column-wise maximum value of $\mathbf{S}^{\mathcal{M}}$;
    Obtain $\mathbf{S}^* \in \mathbb{R}^{[(t-1) \times N_p] \times N_t}$ by column-wise expansion of $\boldsymbol{v}$;
    Difference matrix $\mathbf{S}'$: $\mathbf{S}'_{i,j} = \max\{\mathbf{S}^0_{i,j} - \mathbf{S}^*_{i,j}, 0\}$, Eq. 5;
    Cumulative score histogram $H$: $H_i = \sum_{j=1}^{N_t} \mathbf{S}'_{i,j}$, Eq. 6;
    Top-1 element $k = \arg\max H$;
    Update $\mathcal{M}_t \to \mathcal{M}_t \cup \{(\boldsymbol{p}_k, \boldsymbol{k}_k)\}, (\boldsymbol{p}_k, \boldsymbol{k}_k) \in \mathcal{P}$;
**end for**
Return $\mathcal{M}_t$

---

**Algorithm 2** Aligning-guided New Prompt Learning ($\Psi_{\mathcal{L}}$)

---

**Input:** Data $D_t = \{(\boldsymbol{x}_i, y_i)\}_{i=1}^{N_t}$, Prompt pool $\tilde{\mathcal{P}}_{t-1} = \mathcal{P}_1 \cup \mathcal{P}_2 \cup \cdots \cup \mathcal{P}_{t-1}$
**Output:** New prompt set $\mathcal{P}_t$
Initialize $\mathcal{P}_t = \mathcal{M}_t$;
**for** $m = 1$ **to** $N_{iter}$ **do**
    Simple $(\boldsymbol{x}, y)$ from $D_t$;

    *# New Prompt Training module ($\boldsymbol{f}_T$)*
    Obtain similar scores over $\mathcal{P}_t$: $\{\mathcal{S}(q(\boldsymbol{x}), \boldsymbol{k}_t^i) = (1 + cos(q(\boldsymbol{x}), \boldsymbol{k}_t^i))/2\}_{i=1}^{N_p}$, Eq. 2;
    Obtain top-$K$ scores $\mathcal{S}_{\boldsymbol{x}}^t = \{\dot{\boldsymbol{s}}_t^1, \dot{\boldsymbol{s}}_t^2, \ldots, \dot{\boldsymbol{s}}_t^K\}$;
    Obtain minimum matched score $\alpha = \min \mathcal{S}_{\boldsymbol{x}}^t$;
    Obtain top-$K$ prompts $\mathcal{K}_{\boldsymbol{x}}^t = \{\dot{\boldsymbol{p}}_t^1, \dot{\boldsymbol{p}}_t^2, \ldots, \dot{\boldsymbol{p}}_t^K\}$;
    Obtain compositional prompt $\overline{\boldsymbol{p}}_{\boldsymbol{x}}^t = g_c(\dot{\boldsymbol{p}}_t^1, \dot{\boldsymbol{p}}_t^2, \ldots, \dot{\boldsymbol{p}}_t^K) = \frac{1}{K} \sum_{i=1}^K \dot{\boldsymbol{p}}_t^i$, Eq. 3;
    New data learning loss $\mathcal{L}_{new} = \text{CE}(f_\theta(\boldsymbol{x}, \overline{\boldsymbol{p}}_{\boldsymbol{x}}^t), y)$, Eq. 7;

    *# Historical Prompt Online Aligning module ($\boldsymbol{f}_A$)*
    Obtain similar scores over $\tilde{\mathcal{P}}_{t-1}$: $\{\mathcal{S}(q(\boldsymbol{x}), \boldsymbol{k}_i) = (1 + cos(q(\boldsymbol{x}), \boldsymbol{k}_t^i))/2\}_{i=1}^{N_p \times (t-1)}$, Eq. 2;
    Obtain top-$K$ scores $\mathcal{S}_{\boldsymbol{x}}^{t-1} = \{\dot{\boldsymbol{s}}_{t-1}^1, \dot{\boldsymbol{s}}_{t-1}^2, \ldots, \dot{\boldsymbol{s}}_{t-1}^K\}$;
    Obtain top-$K$ prompts $\mathcal{K}_{\boldsymbol{x}}^{t-1} = \{\dot{\boldsymbol{p}}_t^1, \dot{\boldsymbol{p}}_t^2, \ldots, \dot{\boldsymbol{p}}_t^K\}$;
    Obtain old prompt weight $w_{t-1}^i = e^{\min\{\dot{s}_{t-1}^i - \alpha, 0\}/\tau}$, , Eq. 8;
    Obtain alignment prompt $\overline{\boldsymbol{p}}_{\boldsymbol{x}}^{agn} = \frac{1}{K + \sum_{i=1}^K w_{t-1}^i} \sum_{i=1}^K (w_{t-1}^i \dot{\boldsymbol{p}}_{t-1}^i) + \frac{K}{K + \sum_{i=1}^K w_{t-1}^i} \overline{\boldsymbol{p}}_{\boldsymbol{x}}^t$, Eq. 9;
    Calculate knowledge alignment loss $\mathcal{L}_{agn} = \text{CE}(f_\theta(\boldsymbol{x}, \overline{\boldsymbol{p}}_{\boldsymbol{x}}^{agn}), y)$, Eq. 10;
    Optimize $\mathcal{L} = \mathcal{L}_{new} + \lambda \mathcal{L}_{agn}$, Eq. 11;
**end for**
Return $\mathcal{P}_t$

---

Note that in Alg. 1, at the beginning of the Greedy Prompt Search module, the Reusable Prompt Memory is empty, and the Memory Matrix $\mathbf{S}^M \in \mathbb{R}^{0 \times N_t}$ is an empty matrix. Then, the column–wise max process is conducted to obtain a $1 \times N_t$ null matrix. Consequently, the Assignment Matrix $\mathbf{S}^*$ become a $([(t-1) \times N_p] \times N_t)$ null matrix. Besides, the Cumulative score histogram $H$ could be a zero-vector. This phenomenon arises since the new domain is highly relevant to a small subset of historical prompts, and the remaining historical prompts are considered useless. In such conditions, we randomly select $R = 2$ prompts in $\mathcal{M}_t$ and generate a prompt $(\boldsymbol{p}_k, \boldsymbol{k}_k)$ by through their interpolation. Then $(\boldsymbol{p}_k, \boldsymbol{k}_k)$ is added to $\mathcal{M}_t$.

## B. Visualization Results.

**Seen-Domain Evaluation Curve** In addition to our main paper, we also provide the seen domain performance tendency on the other two benchmarks, *i.e.*, DomainNet and ImageNet-Mix, which are illustrated in Fig. 8 and Fig. 9, respectively. The results show that although our method obtains comparable performance with the existing methods in the initial training stages, it consistently outperforms the competitors after learning from 3-th and 10-th domains on DomainNet and ImageNet-Mix respectively, attributed to our componential prompt-knowledge alignment designs that effectively utilized the learned knowledge to boost both training and inference along the continual learning process.

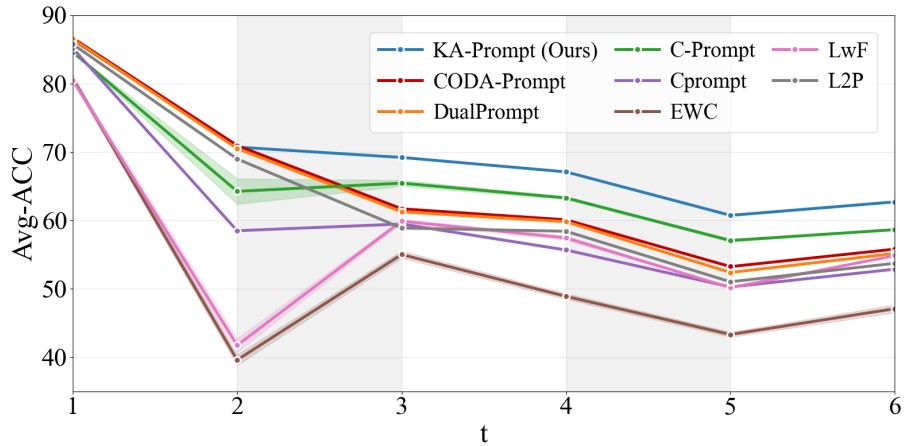

Figure 8: The seen domain performance tendency on DomainNet.

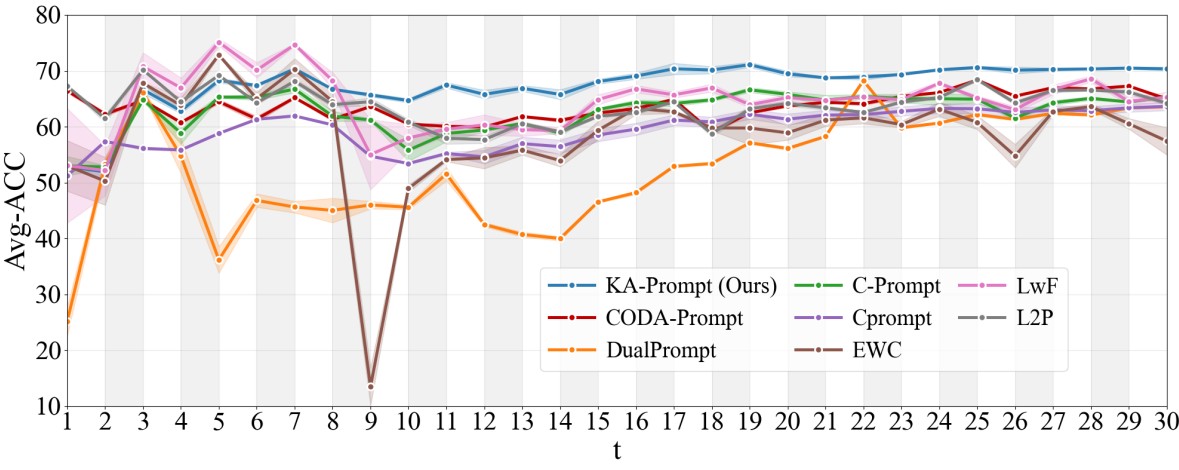

Figure 9: The seen domain performance tendency on ImageNet-Mix.

# C. Parameters and Overhead Comparison

We also compare the model parameters, training overhead, and performance with state-of-the-art prompt-based methods in Tab. 3. All experiments are conducted on the ImageNet-R benchmark. The numbers of total parameters and trainable parameters in KA-Prompt are identical to the baseline C-Prompt (Liu et al., 2024a) since **no extra learnable parameter** is introduced in our method. The number of trainable parameters is larger than other methods, *e.g.*, CPrompt, CODA-Prompt, and DUalPrompt. This is because the competitors learn one long prompt for each domain, while C-Prompt and our KA-Prompt learn a set of short prompts per domain. Our GPU Memory cost and training Batch Time are comparable to most prompt-based methods. The GPU Memory cost and training Batch Time improvement compared to the baseline is due to the Historical Prompt Online Aligning module ($f_A$) which introduces extra computing.

As for inference, our method adopts the same prompt matching, fusion, and utilization process as C-Prompt **without introducing extra operation**. Given the **comparable training and inference overhead**, our method outperforms the existing methods by **4.08%-39.28%**, highlighting the effectiveness and feasibility of our method.

Table 3: Comparison of the number of parameters, overhead and performance with the state-of-the-art on ImageNet-R.

| Methods | Total Params | Trainable Params | GPU Memory (GB) | Batch time (S) | Avg-ACC |
|---|---|---|---|---|---|
| L2P (Wang et al., 2022c) | 86,263,843 | 311,387 | 19.33 | 0.66 | 56.55±0.33 |
| S-Prompts (Wang et al., 2022a) | 92,964,094 | 1,637,910 | 14.90 | 0.39 | 27.23±0.16 |
| DualPrompt (Wang et al., 2022b) | 86,514,211 | 561,755 | 19.45 | 0.66 | 59.47±1.00 |
| CODA-Prompt (Smith et al., 2023) | 88,437,580 | 2,638,926 | 19.18 | 0.80 | 55.21±0.33 |
| CPrompt (Gao et al., 2024) | 92,485,224 | 2,689,168 | 41.64 | 1.18 | 59.48±1.09 |
| C-Prompt (Liu et al., 2024a) | 89,180,505 | 3,381,851 | 11.99 | 0.42 | 62.43±0.49 |
| KA-Prompt (Ours) | 89,180,505 | 3,381,851 | 19.28 | 0.72 | 66.51±0.36 |

