# OpenReview forum: "Componential Prompt-Knowledge Alignment for Domain Incremental Learning"
_ICML.cc/2025/Conference — ICML 2025 poster_

### Official Review · Reviewer_3qyA · 2025-02-16

**Overall Recommendation:** 3

**Summary:**

Domain Incremental Learning (DIL) is crucial for processing data across different domains while maintaining previously acquired knowledge, but current prompt-based methods suffer from misalignment issues when integrating knowledge from different domains. The authors identify that this problem stems from random positioning of knowledge components within prompts, which can lead to interference when irrelevant components are combined. To address this, they introduce KA-Prompt, a novel method that focuses on component-aware prompt-knowledge alignment during the training process. The approach works in two phases: first establishing alignment between new and old prompts through initial structure configuration, and then preserving this alignment through dynamic identification of relevant prompts and adaptive consistency constraints. Through extensive testing on DIL benchmarks, KA-Prompt demonstrates significant improvements over existing methods, showing the effectiveness of their component-aligned approach.

## update after rebuttal

The experiments provided in rebuttal demonstrate that reusable knowledge mining can capture richer and more accurate semantic representations. However, it seems that the observation that “the semantic partial knowledge of objects in the new prompts is continuously reinforced during incremental learning” has not been observed. Moreover, the contribution of the proposed prompt fusion mechanism is ambiguous due to the superiority of individual prompts. Therefore, I decided to keep my score.

**Claims And Evidence:**

Yes. This paper randomly shuffles different components of prompt during prompt fusion, and the resulting fluctuations demonstrate that the previous fusion method did not achieve knowledge alignment.

**Essential References Not Discussed:**

In my understanding, the important references have already been discussed.

**Experimental Designs Or Analyses:**

The authors conducted experiments on four benchmark datasets and compared state-of-the-art cue-based incremental learning methods. The overall experimental design is sound.

**Methods And Evaluation Criteria:**

The proposed method is meaningful for the domain incremental learning it focuses on. The paper provides evidence that KA-Prompt enhances the ability to integrate cross-domain knowledge in continuous learning. However, further analysis is needed regarding the mitigation of knowledge mismatch in the prompt component.

**Other Comments Or Suggestions:**

None.

**Other Strengths And Weaknesses:**

Strengths:

1. The figures and tables in this paper are clearly represented.
2. The paper provides a detailed and clear description of the limitations of the previous work and the motivation for the proposed methodology.

Weaknesses:

1. Insufficient analysis of experiments.
2. Some parts of the proposed method lack definition.

**Questions For Authors:**

I have some concerns about this paper:

1. As shown in Figure 3, KA-Prompt needs to maintain a prompt pool in addition to the prompt set. Is this a requirement specific to KA-prompt or is it already present in the baseline method? The additional training overhead associated with prompt pools should be discussed.
2. Does the aligned prompt get updated during training? If so will these updates have an effect on the prompt parameter in Reusable Prompt Memory?
3. Following up on the previous question, is the purpose of Historical Prompt Online Aligning to align the new prompt to the historical prompt pool, or vice versa?
4. The ablation study does not seem to indicate on which dataset it was performed. Ablation studies on more datasets can further demonstrate the contribution of the proposed module.
5. Can the misalignment illustrated in Figure 2 and the optimization resulting from the proposed method be further analyzed in the form of a heat map visualization?

**Relation To Broader Scientific Literature:**

This paper improves on previous work in two ways:

1. It optimizes the initialization mechanism of the newly introduced prompt. Instead of directly using the prompt obtained from the previous task as the initialisation, a suitable prompt is searched from the existing prompts as the initialisation of the new prompt.

2. Different from previous work that directly merges the components in the corresponding positions in the prompt, this article aligns knowledge before merging, thereby reducing the ineffective merging of unrelated knowledge.

**Theoretical Claims:**

This article does not involve theoretical proofs.

---

> ### Author Rebuttal · Authors · 2025-03-31
>
> Thank you for your constructive feedback and recognition. Below are our responses, which we hope effectively address your concerns.
>
> **Q1-1: Requirement of maintaining a prompt pool.**
>
> (1) Maintaining a prompt pool is not a specific requirement of KA-Prompt but is already present in the baseline method.
>
> (2) A prompt pool is fundamental to prompt-based continual learning methods, enabling the retention of historical knowledge with negligible storage overhead. All compared state-of-the-art prompt-based approaches, including L2P, S-Prompts, DualPrompt, CODA-Prompt, CPrompt, and C-Prompt, incorporate a prompt pool.
>
> **Q1-2: Additional training overhead**
>
> (1) Maintaining a prompt pool does not introduce additional training overhead since stored prompt sets are frozen after being added and are not further optimized.
>
> (2) In our KA-Prompt, frozen prompts from the pool are selected to guide cross-domain componential knowledge alignment in the Historical Prompt Online Aligning (HPOA) branch. This branch shares nearly the same computational cost as the New Prompt Training branch, introducing only an additional forward pass. However, the HPOA branch contributes a **2.28%** improvement when applied to the baseline model on the ImageNet-R benchmark, as shown in Fig. 5 of our main paper.
>
> **Q2: Aligned prompt update**
>
>  (1) The aligned prompts are updated during training to encode knowledge from new domains.
>
>  (2) The prompt parameters in the Reusable Prompt Memory remain frozen during the aligned prompt update, ensuring they are unaffected during training.
>
> **Q3: Purpose of Historical Prompt Online Aligning module**
>
>  The Historical Prompt Online Aligning (HPOA) module aims to align the new prompts to parts of historical prompts in the prompt pool. Specifically, as new prompts are updated, HPOA dynamically matches them to the closest historical prompts, assigns fusion weights, and fuses them. The fused prompts are then fed into the ViT model, ensuring cross-domain componential alignment throughout training.
>
> **Q4: Sblation study**
>
> (1) By default, module ablation studies in the main paper are conducted on ImageNet-R. This has been explicitly stated in the revised version.
>
>  (2) Additional ablation studies on ImageNet-Mix and DomainNet are provided in Fig. G of https://anonymous.4open.science/r/ICML-31/FigG-Ablation.png.
>  Specifically, the results show that:
>
>  (a) Our prompt initialization strategy (Reusable Knowledge Mining, $\boldsymbol{f}_R + \boldsymbol{f}_G$) significantly outperforms the existing methods (Wang et al., 2023a), due to the improved historical knowledge utilization capacity.
>
>  (b) Our online alignment design ($\boldsymbol{f}_A$) consistently improves model performance by strengthening knowledge alignment during training, thereby enhancing cross-domain fusion compatibility at test time.
>
>  (c) When all our modules are used together, the model performance is further improved since the historical knowledge utilization is improved during both training and testing.
>
> **Q5: Heat map visualization**
>
>  Thanks for your suggestion! We have visualized the attention maps of prompt tokens at different stages, along with the fused tokens, in  Fig. A of https://anonymous.4open.science/r/ICML-31/FigA-Heatmap.png. The results show that each prompt token (component) captures a semantic part of objects.
>
> (1) Due to semantic misalignment among tokens, the fused prompt tokens in C-Prompt fail to precisely capture object-specific information, preventing the model from fully leveraging discriminative features.
>
> (2) Our method improves component-wise alignment, enabling the fused prompt token to effectively capture object features. This results in a **4.73%** Average accuracy improvement in Tab. 1 of our main paper.

---

> > ### Comment · Reviewer_3qyA · 2025-04-04
> >
> > Thanks to the author for his reply, which addressed most of my concerns. Regarding the heat map provided by the authors in their rebuttal, in addition to the fused prompts, KA-Prompt is also significantly better on the prompts of individual domains compared to C-Prompt. This seems to naturally lead to better attention performance in the fused prompt. Can this phenomenon be explained and the advantages of the fusion approach proposed in this paper illustrated?

---

> > > ### Author Response · Authors · 2025-04-04
> > >
> > > Thank you for your valuable feedback. We hope the following responses address your concerns:
> > >
> > > **Q1: Advantages in prompt learning for individual domains**
> > >
> > > Our KA-Prompt generates significantly better visualization results for individual domain prompts compared to C-Prompt. The key advantages of our KA-Prompt are as follows:
> > >
> > > (1) C-Prompt learns prompts **independently** across domains, disregarding the accumulation of semantic knowledge throughout continual learning. This limits its ability to retain and leverage previously acquired knowledge during new prompt learning.
> > >
> > > (2) In contrast, our *Reusable Knowledge Mining* mechanism actively incorporates semantic knowledge from previously learned domains into new prompts. As a result, the semantic partial knowledge of objects in the new prompts is **continuously reinforced** during incremental learning. This process enables our learned prompts to capture richer and more precise semantic representations than those of C-Prompt.
> > >
> > >
> > > **Q2: Advantages in prompt fusion**
> > >
> > > Our KA-Prompt achieves more effective cross-domain knowledge utilization during fusion compared to C-Prompt. The key advantages of our KA-Prompt are as follows:
> > >
> > > (1) As demonstrated in our visualization results in Fig. A of https://anonymous.4open.science/r/ICML-31/FigA-Heatmap.png, the prompts of C-Prompt on individual domains can capture object-specific semantic information such as the wing of a fighter jet, the head and tail of an airplane, the cabin of a boat, and the steel cables and deck of a bridge.
> > > However, due to semantic misalignment across different domains, **knowledge conflicts** arise during prompt fusion. As a result, the fused prompt of C-Prompt often fails to precisely capture the semantic regions of objects, leading to limited cross-domain knowledge utilization and degraded performance.
> > >
> > >
> > > (2) In our KA-Prompt, prompt tokens at the same position across different domains encode highly relevant semantic information, significantly improving **knowledge compatibility** during prompt fusion. For instance, in the upper sample of Fig. A(a) of https://anonymous.4open.science/r/ICML-31/FigA-Heatmap.png, Token1, Token2, Token3, and Token4 of different domains primarily encode the body, head, vertical fin, and missile of a fighter jet, respectively. As a result, the fused prompt tokens effectively capture discriminative object regions, enabling more efficient utilization of accumulated semantic knowledge across domains and leading to a **4.73%** average improvement across four DIL benchmarks.

---

### Official Review · Reviewer_AMjb · 2025-02-25

**Overall Recommendation:** 4

**Summary:**

This paper focuses on the domain incremental learning (DIL) task and identifies component-wise misalignment between domain-specific prompts as a key factor that leads to conflicting knowledge integration and degraded predictions in prompt-based DIL methods. To address this issue, the authors propose the Componential Prompt-Knowledge Alignment (KA-Prompt) approach, which introduces a dual-phase framework to enhance component-wise alignment, thereby improving knowledge utilization during both training and inference. Extensive experimental results on four DIL benchmarks demonstrate that KA-Prompt achieves promising improvements compared to state-of-the-art methods.

**Claims And Evidence:**

Yes

**Essential References Not Discussed:**

No

**Experimental Designs Or Analyses:**

Yes. I have checked the experimental designs and analyses in Figures 1, 2, 4, 5, 6, 7, 8, 9, and Tables 1, 2, 3. These experiments are sound and comprehensively demonstrate the motivation, effectiveness, and efficiency of the proposed method.

**Methods And Evaluation Criteria:**

Yes

**Other Comments Or Suggestions:**

Figure 7 is a crucial experiment that verifies the effectiveness of the proposed approach in addressing knowledge misalignment compared to the existing methods. It is recommended that this figure be moved to the main paper.

**Other Strengths And Weaknesses:**

Paper strength:
1. The manuscript is well-organized and clearly written, making complex concepts accessible to a broad audience. The visual representations are well-designed and effectively illustrate the motivation, methodological designs, and effectiveness of the proposed method.
2. A deep understanding of prompt-based DIL technologies is demonstrated through a comprehensive literature review. The identification of component-wise misalignment between domain-specific prompts, supported by well-designed experiments, is insightful and offers inspiration for further research in handling domain shifts.
3. The proposed method is sound and innovative. First, the Greedy Prompt Search module provides a novel, low-cost solution to improve the utilization of historical prompts, which could be highly beneficial to incremental learning and transfer learning communities. Second, the Historical Prompt Online Aligning module dynamically matches historical prompts and adopts an adaptive strategy to constrain the prompt alignment, effectively addressing the componential knowledge misalignment identified in the paper.
4. Extensive experiments on multiple benchmarks are conducted, with the proposed method demonstrating notable improvements in the DIL task. Additionally, sufficient ablation studies are included, verifying that the proposed designs effectively achieve the claimed objectives.

Paper weakness:
1. Some experimental results require further analyses. For instance, in Figure 4(a), KA-Prompt and the C-Prompt baseline underperform Dual-Prompt, CODA-Prompt, and CPrompt in the first domain. The reasons for this phenomenon should be discussed.
2. Certain methodological details are not sufficiently introduced. Specifically, if the Reusable Prompt Memory is empty, the Memory Matrix should also be empty. In this case, does the Assignment Matrix S^* become a ( [(t−1)×N_p]×N_t) null matrix? The authors should clarify how such a case is handled, either in the main paper or in the appendix.
3. The ViT model in Figures 1(b) and 3 should be illustrated with a consistent shape to improve clarity.

**Questions For Authors:**

This paper presents an new DIL approach with extensive evaluations. The presentation and experiments are comprehensive, I have a few minor concerns regarding the experimental analyses and implementation details for special cases. Please address these issues, as outlined in the paper weaknesses section, during the rebuttal.

**Relation To Broader Scientific Literature:**

This paper provides a novel solution to enhance cross-domain knowledge utilization during both testing and sequential training. The proposed ideas have the potential to inspire further research in areas involving domain shifts and incremental learning, such as transfer learning and class/task incremental learning.

**Theoretical Claims:**

This paper does not provide a theoretical claim or proof. However, the challenges identified in existing methods are intuitive and are effectively demonstrated through well-designed experiments.

---

> ### Author Rebuttal · Authors · 2025-03-31
>
> We sincerely appreciate the reviewer’s constructive feedback and recognition. We hope the following responses effectively address your concerns.
>
> **W1: Performance analysis**
>
> (1) Both KA-Prompt and the C-Prompt baseline exhibit lower performance on the initial domain compared to other prompt-based methods due to the BEMA design proposed in C-Prompt. BEMA is a batch-wise antiforgetting strategy that prevents the forgetting of cross-domain shared parameters (e.g., classifier). Since BEMA constrains new knowledge learning, and the randomly initialized shared parameters lack semantic knowledge, it limits the initial performance of both KA-Prompt and C-Prompt, particularly on small-scale datasets like ImageNet-R.
>
> (2) To further analyze this effect, we removed BEMA during the first domain learning, and the results are presented in Fig. E of https://anonymous.4open.science/r/ICML-31/FigE-ImageNet-R.png. The findings show that: (a) Without BEMA, our model achieves performance comparable to state-of-the-art methods on the first domain, confirming that BEMA is the primary factor contributing to the degraded initial performance. (b) After learning from 15 domains, our KA-Prompt (w/o initial BEMA) achieves **66.54±0.68**, which is 0.03% higher than our results in the paper (w/ initial BEMA). This marginal performance improvement occurs because some discriminative knowledge is shared across domains. Thus, even if the initial domain is insufficiently trained, knowledge from later domains can still enhance its performance within our framework. (c) Overall, KA-Prompt achieves **4.08%** and **4.11%** improvements on ImageNet-R with and without initial BEMA, respectively, verifying its effectiveness in consolidating long-term knowledge. These improvements stem from our knowledge alignment design, which enhances cross-domain prompt compatibility, enabling more efficient utilization of learned knowledge and facilitating both positive forward and backward transfer.
>
> **W2: Methodological details**
>
> Your understanding is correct.
>
> (1) At the beginning of Greedy Prompt Search, the Reusable Prompt Memory is empty, and the Memory Matrix $S^M\in \mathbb{R}^{0\times N_t}$ is an empty matrix. Then, the column–wise max process is conducted to obtain a $1\times N_t$ null matrix. Consequently, the Assignment Matrix $S^*$ become a ($[(t−1)×N_p]×N_t$) null matrix.
>
> (2) We have incorporated these methodological details into our paper for improved clarity.
>
> **W3: Unify the component shapes**
>
>  Thank you for this valuable suggestion. We have modified the shape of the ViT model in Fig. 1(b) to match Fig. 3, ensuring overall consistency. The revised version is illustrated in Fig. F in https://anonymous.4open.science/r/ICML-31/FigF-Modify.png.
>
>
> **Comments: Moving Figure 7 to main paper**
>
> Thanks for the valuable suggestions and appreciation of our experimental designs. We have moved Fig. 7 to our main paper. We have moved Fig. 7 to the main paper. Additionally, the following experimental settings and quantitative analyses are included:
>
> To demonstrate that componential prompt-knowledge alignment mitigates knowledge conflicts, we conducted an ablation study by shuffling prompt components under different conditions. The experiments were performed on the final domain of the DomainNet benchmark, where all learned domain prompts were used for prompt matching.
> As shown in Fig. 7 (a), C-Prompt (Liu et al., 2024a) experiences performance improvement in Shuffle-B/C/D/E due to its randomly learned componential structure, which tends to be suboptimal in its original form. In contrast, KA-Prompt consistently degrades when prompt components are perturbed, indicating its intrinsic alignment structure. Despite the misalignment noise introduced by random shuffling, the worst performance of KA-Prompt (Fig. 7 (b) Shuffle-E) remains **6.4%** higher than the best performance of C-Prompt (Fig. 7 (a) Shuffle-B). This demonstrates that our greedy prompt search algorithm effectively extracts generalizable knowledge across domains, significantly enhancing adaptation to new domains.

---

### Official Review · Reviewer_8nTg · 2025-02-27

**Overall Recommendation:** 4

**Summary:**

This paper introduces a novel component-based prompt knowledge alignment method, KA-Prompt, for Domain Incremental Learning (DIL). Its key contribution lies in addressing the cross-domain prompt misalignment problem, which is claimed to be a major limitation of existing prompt-based DIL methods, such as C-Prompt. The proposed framework consists of two core mechanisms: Reusable Knowledge Mining (ΨM), which selects and initializes new prompts based on relevant past knowledge, and Aligning-guided New Prompt Learning (ΨL), which dynamically maintains component alignment across domains. Experimental results on four benchmarks (DomainNet, ImageNet-R, ImageNet-C, ImageNet-Mix) demonstrate that the proposed method outperforms state-of-the-art approaches.

**Claims And Evidence:**

In this paper, the authors propose that changes in the order of prompts lead to significant variations in accuracy, which leads to the issue of prompt misalignment. Although the ablation experiment based on shuffling (Fig. 7) provides some preliminary insights, the paper does not offer sufficient theoretical analysis to explain the importance of prompt order and the negative impact of prompt misalignment on model performance. Therefore, more experiments are needed to further validate the actual impact of prompt misalignment and eliminate potential experimental biases. To strengthen the credibility of this claim, it is recommended that the authors add more experiments, conduct broader validations, and quantify the specific effects of prompt misalignment on model performance.

**Essential References Not Discussed:**

The authors discuss and compare a wide range of related methods.

**Experimental Designs Or Analyses:**

The experimental design of this paper aims to verify the effectiveness of KA-Prompt in Domain Incremental Learning (DIL) tasks through comparisons with multiple baseline methods, ablation studies, and hyperparameter analyses. Overall, the experimental design is relatively comprehensive. However, there are some potential shortcomings that require further investigation: The motivation section suggests that prompt misalignment is a major performance bottleneck of C-Prompt, but the ablation study (Fig. 7) only preliminarily validates this hypothesis by shuffling prompt components, without further quantifying the specific impact of misalignment on model representations. Additionally, the scale of the experiments is not sufficient to rule out potential experimental biases.

**Methods And Evaluation Criteria:**

The proposed method and evaluation criteria, including the use of benchmark datasets like DomainNet and ImageNet-Mix, are meaningful for addressing the challenges in Domain Incremental Learning (DIL) and enhancing cross-domain knowledge transfer.

**Other Comments Or Suggestions:**

Please refer to the weaknesses.

**Other Strengths And Weaknesses:**

One of the main contributions of this paper is to explicitly identify component-level misalignment as a key issue in prompt-based Domain Incremental Learning (DIL). Previous works, such as C-Prompt and CODA-Prompt, primarily focused on selecting and fusing relevant prompts but largely overlooked the potential knowledge interference caused by the random positioning of prompt components across different domains.

The main weaknesses of this paper are as follows:

1. Overall, the core contribution of this paper lies in optimizing C-Prompt rather than proposing an entirely new framework, making it more of an incremental improvement with limited novelty. The primary contribution is addressing the prompt misalignment issue in cross-domain learning to enhance prompt fusion across different domains. However, it remains based on the existing C-Prompt structure, with its methodological innovation mainly reflected in localized improvements rather than introducing a new paradigm for incremental learning.

2. The authors propose that different prompt orders lead to significant variations in accuracy, thereby introducing the issue of prompt misalignment. While the shuffle-based ablation study (Fig. 7) provides some preliminary insights, the paper lacks sufficient theoretical analysis to explain the importance of prompt order and the negative effects of prompt misalignment. More experiments are needed to further validate the actual impact of prompt misalignment and rule out potential experimental biases. To enhance the credibility of this claim, the authors are encouraged to conduct broader experiments and quantify the specific impact of prompt misalignment on model performance.

3. Additionally, the paper does not provide a formal analysis or complexity evaluation of the greedy search for reusable knowledge (ΨM). Since greedy algorithms often lead to suboptimal solutions, analyzing its approximation guarantees would strengthen the theoretical depth of the proposed method.

**Questions For Authors:**

1.What constitutes a "component" within a prompt? How is semantic consistency ensured across components in different domains? If the definition of components within prompts is vague or semantically unclear (e.g., if the components are simply vector representations without strict semantic binding), then the effectiveness of the proposed component-level alignment may face fundamental issues.

2.Does component alignment really alleviate conflicts? The paper assumes that aligning prompt components will inevitably reduce knowledge conflicts; however, this assumption has flaws: does the alignment of components necessarily lead to alignment at the semantic or knowledge level? Is it possible that structural alignment does not result in actual semantic alignment?

3.The authors did not provide sufficient qualitative or visual evidence to demonstrate that component-level alignment achieves the expected semantic fusion effect.

4.How does the greedy search ensure that the selected set of prompts is truly globally optimal? The paper fails to provide enough evidence to verify whether such locally optimal solutions are sufficient, or whether they might lead to unstable performance or degradation in large-scale, multi-domain conditions. The paper does not explore whether using a greedy algorithm could introduce significant performance fluctuations, nor does it provide comparisons with other heuristic algorithms to justify the choice of a greedy approach.

5.Could the fusion of multiple prompts cause negative transfer across domains? In cases where there are significant differences in prompt knowledge between domains, does this fusion always lead to positive gains? The paper lacks an analysis and discussion of how the fusion mechanism might trigger negative transfer.

**Relation To Broader Scientific Literature:**

The core contribution of this paper lies in optimizing C-Prompt, with the main contribution being the resolution of prompt misalignment in cross-domain learning to enhance cross-domain prompt fusion. C-Prompt (Compositional Prompting) is a prompt-based Domain Incremental Learning (DIL) method that aims to adapt to cross-domain tasks by learning compositional prompts.

**Theoretical Claims:**

No, the paper does not involve theoretical claim.

---

> ### Author Rebuttal · Authors · 2025-03-31
>
> Thanks for the valuable feedback and comments. We hope the following responses address your concerns.
>
> **W1: Contributions on framework**
>
> The C-Prompt baseline corresponds to our New Prompt Training branch. We have made two key designs to form a brand-new DIL framework:
>
> (1) Reusable Knowledge Mining (RKM) mechanism. Unlike existing methods that randomly initialize new prompts, RKM actively searches for old prompts containing shared knowledge between the new and all old domains, significantly improving new domain adaptation and cross-stage knowledge alignment.
>
> (2) Historical Prompt Online Aligning (HPOA) branch. HPOA introduces an online search and re-weighting based prompt fusion strategy to mitigate cross-stage knowledge drift during training, effectively improving the utilization of multi-domain knowledge.
>
> **W2: Quantization of misalignment**
>
> To quantify misalignment, we have traversed the cross-stage prompt token orders to obtain the performance of the optimal alignment. As shown in Fig. 7, C-Prompt exhibits a 0.64% degradation compared to the optimal alignment, indicating suboptimal order learning. In contrast, our method achieves **0.57-0.79%** higher performance than alternative orderings, verifying that it successfully attains optimal alignment.
>
> **W3, Q4: Complexity evaluation and theoretical analyses**
>
> The objective of RKM module at stage $t$ is formulated as follows: given $m$ training samples and $n=(t-1)×L_s$ old prompts, obtain $k=L_s$ old prompts that minimize $\sum_{i=1}^{m}\{\max_{1≤j≤n} s(x_i,p_j)\}$ where $s(x_i,p_j)$ denotes the similarity between a training sample $x_i$ and a prmpt $p_j$.
>
> (1) The complexity of our greedy search algorithm is $mnk=O(nk)$, more efficient than exhaustive grid search, which requires $mC_n^k=O(n^k)$ operations.
>
> (2) The optimization of RKM can be approximated as a k-medoids problem [1] by considering old prompts as special training samples (since m>>n, this approximation does not affect the theoretical conclusion). According to Theorem 4.4 of [1], in a one-way search setting, the error bound of our solution $E(θ^*)$ relative to the optimal solution $E(θ)$ satisfies:
> $E(θ^*)≤(1+\frac{2k}{n+m})E(θ)$. Since m+n>>k, our greedy search ensures globally optimal solutions under different conditions and leads to stable performance.
>
> [1] PAMAE: Parallel k-Medoids clustering with high accuracy and efficiency. SIGKDD, 2017.
>
> (3) We have conducted experiments on ImageNet-R to compare our method with existing approaches:
>
> |Method|Previous-Domain [a]|Most-Similar [b]|Greedy (Ours)|
> |-|-|-|-|
> |Avg-ACC|63.55±0.46|64.32±0.48|**65.36**±0.52|
>
> [a] initializes new prompts using those from the previous domain.
>
> [b] selects prompts with the highest similarity scores in a single search step for initialization.
>
> The results show that our greedy search-based reusable knowledge mining strategy (Greedy) consistently outperforms [a] and [b] with improvements of **1.81%** and **1.04%**, respectively.
>
> Both [a] and [b] suffer from insufficient utilization of old knowledge due to limited knowledge relevance within adjacent domains and overlooking of knowledge in lower-similarity prompts, respectively.
>
> In contrast, our approach evaluates the unique knowledge that has not been included in the selected prompts iteratively, effectively improving the utilization of old knowledge.
>
> **Q1: Definition of components and semantic consistency**
>
> (1) The components refer to the tokens of each prompt. As shown in Fig. 2, each prompt, e.g., $p_t^1$, contains 4 tokens (i.e., components), each encoding a distinct aspect of object knowledge.
>
> (2) In DIL, the categories of different domains are identical, thus the object parts are semantically consistent across domains.
>
> **Q2, Q3: Do alignment alleviate conflicts**
>
> We have visualized the attention maps of prompt tokens at different stages, along with the fused tokens, in  Fig. A of https://anonymous.4open.science/r/ICML-31/FigA-Heatmap.png. The results show that each prompt token (component) captures a semantic part of objects.
>
> (1) Due to semantic misalignment among cross-domain prompt tokens, the fused prompt tokens in C-Prompt fail to precisely capture object-specific information.
>
> (2) Our method improves component-wise alignment, enabling the fused prompt token to effectively encode discriminative features of objects. This results in a **4.73%** increase in Average accuracy, as reported in Tab. 1.
>
> **Q5: Influence of prompt fusion**
>
> Fig. D in https://anonymous.4open.science/r/ICML-31/FigD-Pos-Transfer.png shows the testing performance trends for each domain. The results indicate that 2 out of 14 old domains exhibit performance reduction after continual training. This arises due to data imbalance, where some domains contain a few training samples, leading to biased knowledge during fusion. Nevertheless, our method achieves significant improvements on 12 out of 14 old domains, indicating that it effectively achieves positive knowledge transfer.

---

> > ### Comment · Reviewer_8nTg · 2025-04-02
> >
> > Thank the authors for the exhaustive reply. After carefully reviewing the authors' rebuttal, all my concerns have been sufficiently addressed, including the novelty, claimed problem, theoretical guarantee, and effectiveness of the proposed approach. Overall, this paper focuses on the practical domain incremental learning task and discovers the existence of the prompt misalignment problem. Then, an effective approach, KA-Prompt, is proposed to address the claimed problem. Abundant quantitative, qualitative, and theoretical results and analyses are provided to demonstrate the significance of the proposed approach. Consequently, I am willing to raise my score.

---

> > > ### Author Response · Authors · 2025-04-03
> > >
> > > Dear reviewer 8nTg
> > >
> > > We sincerely appreciate your thoughtful feedback and the time you dedicated to reviewing our work. Your insightful comments have been invaluable in refining our presentation and strengthening the manuscript. We are grateful for the opportunity to clarify our approach and truly appreciate your recognition of our work.
> > >
> > > Best regards,
> > >
> > > Authors

---

### Official Review · Reviewer_W2NY · 2025-03-13

**Overall Recommendation:** 2

**Summary:**

The paper addresses the challenge of DIL. The authors identify a limitation in existing prompt-based methods: component-wise misalignment between domain-specific prompts leads to conflicting knowledge integration and degraded predictions. To address this, they propose ​KA-Prompt, a method that enforces component-wise knowledge alignment across domains. KA-Prompt operates in two phases: (1) ​Initial Componential Structure Configuring, where a set of old prompts containing relevant knowledge is mined via greedy search to initialize new prompts, ensuring reusable knowledge transfer and intrinsic alignment; and (2) ​Online Alignment Preservation, which dynamically identifies target old prompts and applies adaptive componential consistency constraints as new prompts evolve.

**Claims And Evidence:**

No. Please refer to "other strengths and weaknesses" for detail.

**Essential References Not Discussed:**

No.

**Experimental Designs Or Analyses:**

Yes. Please refer to "other strengths and weaknesses" for detail.

**Methods And Evaluation Criteria:**

No. Please refer to "other strengths and weaknesses" for detail.

**Other Comments Or Suggestions:**

Figure 2 is somewhat overly cluttered, making it difficult to grasp the key points.

**Other Strengths And Weaknesses:**

Strengths

1. Extensive experiments on multiple benchmarks demonstrate the superiority of KA-Prompt over state-of-the-art methods.
2. Although I am not entirely convinced by the motivation behind this paper, the issue of how to share knowledge across different domains is indeed a critical problem in DIL. This paper offers a new approach to addressing this challenge.

Weakness:

1. The authors claim in their contributions that "We reveal that component-wise misalignment in prompts limits their cross-domain knowledge integration and utilization capacity." However, the paper does not provide any experimental or theoretical evidence to support this claim. Attention mechanisms in Transformers are inherently permutation-invariant, and methods like Visual Prompt Tuning (VPT) can operate without positional encoding. Therefore, it is unclear why misalignment would occur in the first place. The authors need to provide a more rigorous justification for this claim.
2. The motivation for the paper, as illustrated in Figure 2, is based on two assumptions: (1) different components typically encode distinct types of knowledge, and (2) independently learned prompts exhibit misalignment in componential knowledge, leading to the fusion of irrelevant knowledge during inference. However, these assumptions are not supported by any experiments or references to prior work. Specifically, the depiction of different tokens representing different parts of an airplane in Figure 2 is confusing and lacks empirical validation.
3. The necessity of the Greedy Prompt Search module is questionable. As I understand it, this module computes the similarity between training samples and all prompt keys, then selects the most similar prompts for initialization. Even if the similarity is computed across all training samples, the computational cost would still be less than performing a single forward pass on the training set. The authors should justify the need for this module more clearly.
4. The paper does not adequately address the issue of catastrophic forgetting in DIL. In fact, the authors' approach may increase the risk of forgetting. By using prompts from old tasks to initialize new tasks and encouraging similarity between new and old prompts during Historical Prompt Online Aligning, the model may incorrectly select new prompts for old task data during inference, exacerbating forgetting.
5. The experimental setup is not clearly defined, particularly in Equation 12. It is unclear whether $a_{T,i}$ is evaluated on the test data of the i-th domain only or on the test data of all previous domains. This ambiguity needs to be clarified to ensure the reproducibility and validity of the results.

**Questions For Authors:**

1. Compared to C-Prompt, the motivation for this paper is based on the concept of "misalignment." How is misalignment defined, and how is alignment measured and evaluated? What metrics or experiments are used to determine whether alignment has been achieved?
2. How is the final classifier set up? Is there a shared classifier across all domains, or does each domain have its own separate classifier? This distinction is crucial for understanding the model's ability to generalize across domains.
3. Given the weaknesses identified, how does the proposed method address the issue of catastrophic forgetting in incremental learning? Specifically, how does the method ensure that knowledge from old tasks is not overwritten or forgotten when learning new tasks?
4. Are the prompts from old tasks updated during the Online Aligning process? If so, how does the method ensure that the updated prompts perform better on old tasks compared to the original prompts?
5. Why are the results for S-Prompt missing in Figures 8 and 9? Based on my experimental experience, S-Prompt is effective in reducing forgetting, yet it is notably absent from these comparative results. Could the authors explain this omission and provide the missing results?

**Relation To Broader Scientific Literature:**

The authors effectively build on prior work in prompt-based learning, such as C-Prompt and CODA-Prompt, while addressing a critical limitation (component-wise misalignment) that has not been previously explored in depth.

**Theoretical Claims:**

No proofs.

---

> ### Author Rebuttal · Authors · 2025-03-31
>
> Thanks for the valuable feedback. We hope our responses address your concerns.
>
> **W1: Misalignment's occurring and definition**
>
> (1) Misalignment occurs during cross-stage prompt fusion in DIL. In Fig. 2, each prompt (e.g., $p_t^1$) consists of 4 tokens, each encoding distinct partial knowledge of objects. *Misalignment* refers to the disorder of partial knowledge within prompt tokens of different stages. When fusing prompts from multiple stages, this cross-stage token-level misalignment introduces semantic conflicts in the fused prompt. Then, the sub-optimal fused prompt is fed to the Attention layer, leading to degraded performance.
>
> (2) Misalignment does not occur in VPT because it is designed for static training data, where all prompts are optimized jointly rather than incrementally.
>
> (3) Our method explicitly enhances cross-stage prompt alignment, ensuring that the fused prompt retains a coherent representation of semantics. These high-quality fused prompts thereby lead to improved test performance.
>
> **Q1: Measuring alignment**
>
> (1) To measure alignment, we have traversed different token orders across stages to identify the configuration that yields the highest model performance, which we define as the optimal alignment, as shown in Fig. 7.
>
> (2) The results show that the learned prompt token order in C-Prompt baseline exhibits a performance degradation of 0.64% compared to the optimal alignment, verifying the presence of misalignment.
>
> (3) In contrast, the prompt token order learned by us consistently outperforms alternative orders by 0.57–0.79%, demonstrating that our approach effectively achieves optimal alignment.
>
>
> **W2: Validation of misalignment**
>
> We have visualized the attention maps of prompt tokens at different stages, along with the fused tokens, in  Fig. A of https://anonymous.4open.science/r/ICML-31/FigA-Heatmap.png. The results show that each domain-specific prompt token learns a semantic part of objects. Compared to C-Prompt which introduces semantic misalignment in prompt tokens from distinct domains, the fused prompt tokens in our method effectively preserve part-level information, enabling the model to fully exploit discriminative features.
>
> **W3: Necessity of the Greedy Prompt Search (GPS)**
>
> (1) GPS is necessary because (a) the shared knowledge between new and old domains is distributed across various old prompts, and (b) the knowledge of some old prompts overlaps significantly. A naive selection based solely on high similarity scores in a single search would often lead to redundant prompt selection while overlooking relevant knowledge present in lower-similarity prompts. This results in insufficient utilization of old knowledge. Please refer to Reviewer **8nTg-W3, Q4** for more experimental analyses.
>
> (2) The computational cost of GPS is significantly less than performing a single forward pass on the training set since it primarily involves simple matrix addition and subtraction.
>
> **W4,Q3,Q4: Catastrophic forgetting**
>
> Our method inherits the anti-forgetting capacity of prompt learning and does not significantly risk forgetting:
>
> (1) In our Historical Prompt Online Aligning (HPOA) module, old prompts are frozen, ensuring that knowledge from previous tasks is not overwritten or forgotten.
>
> (2) The prompt selection for old tasks is minimally influenced by HPOA. This is because prompt selection relies on prompt keys, and new prompt keys are only trained by minimizing their distance to new data features.
>
> (3) In Tab. 2, domains are trained sequentially from left to right. Our KA-Prompt performs 1.04% inferiorly to the C-Prompt baseline, while we outperform the C-Prompt from the second domain and obtain **4.25%** Average accuracy improvement across all domains. These results show that our approach achieves a better balance between acquisition and forgetting.
>
> **W5: Metrics setup**
>
> $a_{T,i}$ is evaluated on the test data of the i-th domain. Eq. 12 measures the final model’s performance across all previous domains by computing the average performance over them. We have carefully clarified these details in the revised version.
>
> **Q2: Classifier setup**
>
> The final classifier is shared across all domains, consistent with most prompt-based methods. We have clarified this in the revised version.
>
> **Q5: Visualization of S-Prompt**
>
> (1) To maintain consistency in the color-method pairing relations across Fig. 4 (a)(b), Fig. 8, and Fig. 9, only top-8 methods ranked by Average accuracy are chosen for performance visualization, where S-Prompt is not included.
>
> (2) In Fig. B-1, and Fig. B-2 of https://anonymous.4open.science/r/ICML-31/FigB-S-Prompts.png, we have added the performance curves of S-Prompt. The results show that our KA-Prompt effectively outperforms existing methods during long-term learning.
>
> **Comments: Simplify Fig. 2**
>
> To highlight the misalignment phenomenon, a simplified illustration of Fig. 2 is shown in Fig. C of https://anonymous.4open.science/r/ICML-31/FigC-Simplify.png.

---

> > ### Comment · Reviewer_W2NY · 2025-04-05
> >
> > Thank you for the author's response, which has addressed some of my concerns.
> > - Regarding "Each encoding distinct partial knowledge of objects"​: What evidence supports this claim?
> >
> > - About forgetting: The paper states that "New prompt keys are only trained by minimizing their distance to new data features," but there is no mechanism to ensure that old task data does not become closer to the new keys.
> > ​
> > - On the shared classifier: The paper mentions "The final classifier is shared across all domains." Could you elaborate on this? For example, if each domain has C classes, is the shared classifier a single C-class classifier, or a C×D classifier (where D is the number of tasks)?
> > ​
> > - Regarding S-Prompt's surprisingly low performance: Before the authors pointed it out, I hadn’t noticed that the reported performance of S-Prompt in the paper was so low—contrary to common expectations. I previously tested S-Prompt’s official code (https://github.com/iamwangyabin/S-Prompts) on DomainNet using an ImageNet-1K pretrained model with shallow VPT, achieving around 50% accuracy easily. And  Table 2 reports only ​8% accuracy on Quickdraw, which is highly counterintuitive. Since the paper uses a stronger pretrained model and likely deeper VPT, performance should theoretically be better. Due to the lack of released code, I have doubts about the reliability of the experiments. Could the authors explain why the reported performance is worse?
> >
> > - Additionally, the number of trainable parameters for S-Prompt in the appendix seems unreasonable. Based on my understanding, S-Prompt should only train task-specific prompts and classifiers, so the parameter count should not be that high.
> >
> > - ​Since CDDB is used as a domain incremental dataset in S-Prompt, does these paper also report comparative results on CDDB?

---

> > > ### Author Response · Authors · 2025-04-08
> > >
> > > Thank you for your thoughtful feedback. We sincerely appreciate the opportunity to address your concerns.
> > >
> > > **Q1: Prompt knowledge**
> > >
> > > (1) As illustrated in Fig. A of https://anonymous.4open.science/r/ICML-31/FigA-Heatmap.png, the prompts of both KA-Prompt and C-Prompt are capable of capturing object-part semantic information. For example, prompts are sensitive to object parts such as the wing and head of a fighter jet, the head and tail of an airplane, the body and cabin of a boat, and the steel cables and deck of a bridge. These observations suggest that different prompt tokens focus on distinct object parts. Furthermore, compared to C-Prompt, our KA-Prompt exhibits stronger partial knowledge encoding in Fig. A, attributed to our knowledge alignment design which incrementally accumulates and reinforces semantic representations across domains.
> > >
> > > (2) From the perspective of the attention mechanism, each image token corresponds to a object part , while prompt tokens exhibit varying degrees of similarity to these image tokens. Then, prompt-token pairs with higher semantic relevance are assigned higher attention weights. Consequently, prompt tokens have a substantial influence on the representation of relevant object parts, supporting the claim that they encode distinct partial knowledge.
> > >
> > >
> > > **Q2: Distance between old data and new prompts**
> > >
> > >  Thank you for the insightful suggestion! We have evaluated the prompt matching accuracies on old domains after training on the final domain. Due to time constraints, we conducted experiments on two smaller benchmarks, ImageNet-R and ImageNet-C, each containing 15 domains. Matching accuracy was computed on the first 14 domains after training was completed on the 15th.
> > >  The results below show that KA-Prompt outperforms C-Prompt by **0.1%** and **2.44%**, respectively, verifying that our method effectively avoids mismatching.
> > >
> > > |Benchmark|ImageNet-R|ImageNet-C|
> > > |-|-|-|
> > > |C-Prompt|36.03±0.37|78.18±0.52|
> > > |KA-Prompt|**36.13**±0.18|**80.62**±0.22|
> > >
> > > It is true that no extra constraints on the distances between old data and new prompts keys are introduced in our KA-Prompt compared to the C-Prompt baseline. This old data matching accuracy improvement is attributed to the following:
> > >
> > > (1) In baseline methods like C-Prompt, prompt keys of different domains are typically initialized from a common random distribution. Therefore, the existing methods adopts an **common-to-specific** learning procedure during prompt keys training across domains. However, since the common initial distribution can be significantly distinct from the domain-specific distribution, these methods suffers form unstable training and under-convergence, increasing the risk of mismatched prompt selection during inference.
> > >
> > > (2) In contrast, although our KA-Prompt also adopts a **common-to-specific** learning paradigm, the initialization of new prompt keys is extracted from the most semantically similar keys from prior domains. Thus, the prompt learning is easier and exhibits improved stability, yielding higher inter-domain discriminability. Consequently, prompt matching accuracy on old data improves even without additional constraints.
> > >
> > > **Q3: Classifier**
> > >
> > > Our classifier follows the setting of the C-Prompt baseline. Specifically, if each domain has $C$ classes, the shared classifier is a single $C$-class classifier.
> > >
> > > **Q4: S-Prompt**
> > >
> > > (1) The results of S-Prompt was reproduced about four months ago. Notably, the official codebase of S-Prompt provides only the executable version of S-liPrompt, which incorporates additional language guidance. When switching  the network setting from *slip* to *sip*, we encountered a runtime error: *TypeError: resolve_pretrained_cfg() got an unexpected keyword argument 'kwargs'*. This issue has also been reported by other researchers, but, to our knowledge, no official solution has been provided.
> > >
> > > (2) To proceed, we modified the environment and dependencies to generate the results shown in our paper. It is possible that some hidden parameter mismatches contributed to the degraded performance. Nevertheless, according to the officially reported results of both S-Prompt and C-Prompt, our approach consistently achieves state-of-the-art performance. Besides, note that the other compared methods in this paper are executable by following the official instructions.
> > >
> > > **Q5: Parameters**
> > >
> > > The reported number of trainable parameters for S-Prompt in our paper follows the statistics provided in the official C-Prompt paper. Specifically, it reflects the cumulative number of trainable parameters across 15 domains (including prompts and classification heads). In contrast, many prior works only report per-domain statistics of prompts, which may have caused confusion.
> > >
> > > **Q6: CDDB**
> > >
> > > When evaluating on the deepfake DIL benchmark, CDDB, our KA-Prompt achieves **75.58%** Average Accuracy, outperforming S-Prompt (74.51%) by **1.07%**. This result demonstrates KA-Prompt’s adaptability to real-world DIL settings.

---

### Decision · Program_Chairs · 2025-05-01

**Decision:**

Accept (poster)

**Comment:**

This paper received mixed reviews: three positive, one negative.
It presents the Componential Prompt-Knowledge Alignment (KA-Prompt) for Domain Incremental Learning (DIL), aiming to address the knowledge conflicts and interference caused by the misalignment of components between domain-specific prompts. The positive reviews praise the methodology’s feasibility and novelty, the insightful perspective on prompt-based DIL researches, and the clear, and well-organized writing. The authors conduct a sufficient ablation study to verify the performance improvement achieved by the proposed designs. However, concerns were raised about the lack of experimental support for the motivation and contribution of this paper beyond accuracy. The reviewers urged the authors to explain how “alignment” was measured and evaluated. Detailed settings for classifiers and a comparison with the important SOTA method (S-Prompt) also need to be clarified.
The authors made significant efforts in their response, the Area Chair (AC) feels that the idea of identifying the component-wise misalignment between domain-specific prompts is interesting and inspiring for DIL. The AC recommends accepting the paper, but the concerns raised in the comments should be addressed when preparing the final version.